# LEARNING TO DISCRETIZE DENOISING DIFFUSION ODES

**Vinh Tong**[1,2], **Trung-Dung Hoang**[4], **Anji Liu**[1,3], **Guy Van den Broeck**[3], **Mathias Niepert**[1,2]

[1]University of Stuttgart, [2]IMPRS-IS, [3]UCLA, [4]University of Bern
`vinh.tong@ki.uni-stuttgart.de`

## ABSTRACT

Diffusion Probabilistic Models (DPMs) are generative models showing competitive performance in various domains, including image synthesis and 3D point cloud generation. Sampling from pre-trained DPMs involves multiple neural function evaluations (NFEs) to transform Gaussian noise samples into images, resulting in higher computational costs compared to single-step generative models such as GANs or VAEs. Therefore, reducing the number of NFEs while preserving generation quality is crucial. To address this, we propose LD3, a lightweight framework designed to learn the optimal time discretization for sampling. LD3 can be combined with various samplers and consistently improves generation quality without having to retrain resource-intensive neural networks. We demonstrate analytically and empirically that LD3 improves sampling efficiency with much less computational overhead. We evaluate our method with extensive experiments on 7 pre-trained models, covering unconditional and conditional sampling in both pixel-space and latent-space DPMs. We achieve FIDs of 2.38 (10 NFE), and 2.27 (10 NFE) on unconditional CIFAR10 and AFHQv2 in 5-10 minutes of training. LD3 offers an efficient approach to sampling from pre-trained diffusion models. Code is available at https://github.com/vinhsuhi/LD3.

## 1 INTRODUCTION

Diffusion Probabilistic Models (DPMs) have emerged as a popular class of generative models, demonstrating competitive performance across various tasks, including image synthesis (Ho et al., 2020; Song et al., 2020b; Dhariwal & Nichol, 2021), text-to-image generation (Nichol et al., 2021; Rombach et al., 2022; Gu et al., 2022), 3D point cloud generation (Luo & Hu, 2021), and molecular generation (Hoogeboom et al., 2022). DPMs learn a multi-step transformation from random (e.g., multivariate Gaussian) noise to the data distribution. While they achieve improved sample quality and diversity compared to single-step generative models like GANs (Goodfellow et al., 2014) or VAEs (Kingma et al., 2021), their multi-step nature incurs significant computational overhead.

Distillation-based methods are one category of approaches for accelerating DPMs (Meng et al., 2023; Liu et al., 2022b; Salimans & Ho, 2022; Song et al., 2023). These methods refine the denoising network to improve generation quality while reducing the number of sampling steps. Although these approaches can significantly enhance quality, they require costly re-training or fine-tuning of the entire network. Moreover, distillation-based methods often face challenges such as information loss (Zheng et al., 2024) and difficulties with conditional sampling (Meng et al., 2023).

The second set of approaches capitalizes on the ability to sample from DPMs by solving a corresponding diffusion Ordinary Differential Equation (ODE) (Song et al., 2020a; Lu et al., 2022a; Zhang & Chen, 2022; Liu et al., 2022a; Song et al., 2020a; Lu et al., 2022b; Zhao et al., 2023; Zheng et al., 2024). These methods use more sophisticated numerical solvers for diffusion ODEs and require no neural network retraining. Solving diffusion ODEs involves a mandatory step of selecting discretization time steps, and the choice of discretization points greatly influences sample quality. However, there does not exist a time discretization strategy that works well for all dataset-model-ODE solver combinations. Here, we propose an effective yet lightweight approach to *learn good discretization time steps for any pre-trained DPM.*

Assuming the ODE solving pipeline is differentiable, which holds for most ODE solvers currently in use, our idea is to directly minimize the global truncation error to optimize the time steps, which are input to the solver and hence can be treated as trainable parameters. Specifically, we employ a teacher ODE solver that takes small step sizes to approximate the gold standard solution of the ODE. The student with learnable discretized time steps is then tasked to mimic the teacher's final output given the same input (i.e., the initial condition of the ODE).

One problem with this learning framework is the limited capacity of the student model, which can only optimize a few parameters to mimic a stronger teacher, sometimes leading to "underfitting" and suboptimal performance. To address this, we propose a surrogate objective that is easier for the student to optimize. We validate this objective by theoretically proving its "closeness" to the original distillation objective, upper bounding the KL divergence between the distributions induced by the teacher and the student solvers. The resulting algorithm, termed **L**earning to **D**iscretize **D**enoising **D**iffusion ODEs (LD3), efficiently learns the time discretization by backpropagating through the ODE-solving procedure using the proposed surrogate loss. LD3 can be viewed as an additional step to further improve the sample quality of DPMs after making other design choices such as distilling the denoising model and choosing an ODE solver. Additionally, LD3 is efficient and only requires a small set of random samples from a tractable noise distribution. Table 1 summarizes the benefits that LD3 brings compared to related approaches.

Empirically, we test LD3 on both pixel space (Karras et al., 2022) and latent space DPMs (Rombach et al., 2022) across various resolutions, including 32×32 (e.g., CIFAR10 (Krizhevsky & Hinton, 2009)), 256×256 (e.g., LSUN-Bedroom (Yu et al., 2015), ImageNet (Russakovsky et al., 2015)), and 512×512 text-to-image generation (Rombach et al., 2022; Liu et al., 2023) as well as various types of conditions (e.g., class, text prompt) (Rombach et al., 2022). LD3 performs significantly better than existing time discretization heuristics, especially with small NFE (below 10). Additionally, LD3 can be trained in 5 to 40 minutes on a single GPU.

## 2 RELATED WORK

Table 1: Comparison of LD3 with existing methods, including Watson et al. (2022), AYS (Sabour et al., 2024), GITS (Chen et al., 2024), and DMN (Xue et al., 2024), based on several criteria. The first criterion is training speed: methods that optimize time discretizations in less than one hour are classified as "Fast", while others are considered "Slow". Training stability is true for methods not requiring variance reduction techniques or early stopping to exhibit stable training. A method is, respectively, solver-adaptable and network-adaptable if it considers information about the solver and trained neural networks when optimizing the time steps. Finally, we examine whether a model can directly optimize the global truncation error.

| Criterion | Watson et al. (2022) | AYS | GITS | DMN | LD3 (ours) |
|---|---|---|---|---|---|
| Training speed | Slow | Slow | Fast | Fast | Fast |
| Training stability | ✗ | ✗ | ✓ | ✓ | ✓ |
| Solver adaptability | ✓ | ✓ | ✗ | ✗ | ✓ |
| Network adaptability | ✓ | ✓ | ✓ | ✗ | ✓ |
| Global error optimization | ✗ | ✗ | ✗ | ✗ | ✓ |

A well-established approach to accelerating DPMs involves distilling high-quality denoising networks, which typically require many function evaluations, into models that perform the task in fewer steps with minimal performance loss (Meng et al., 2023; Salimans & Ho, 2022; Fan & Lee, 2023). However, this method requires expensive training (Liu et al., 2022b; Luhman & Luhman, 2021; Meng et al., 2023; Song et al., 2023) before the models can be used for sampling. In comparison, distillation-based methods are significantly slower than our approach—by several orders of magnitude—and lack the flexibility to be used in a plug-and-play manner.

In addition to fine-tuning the denoising network, many methods focus on developing more effective ODE solvers (Lu et al., 2022a; Zhang & Chen, 2022; Liu et al., 2022a; Song et al., 2020a; Lu et al., 2022b; Zhao et al., 2023). The key insight is that truncation errors accumulating during the backward sampling process can significantly degrade the quality of generated images, especially when NFE is reduced. To mitigate this, advanced ODE solvers are required. However, since these solvers still rely

on multi-step sampling, selecting an appropriate strategy is crucial. Current approaches often rely on handcrafted schedules, which may not be optimal.

Recent work has focused on optimizing time schedules. Xue et al. (2024) formulate an optimization problem aimed at identifying the optimal time discretization. They derive an upper bound for the global truncation error under the assumption that the score prediction error of the pretrained model is uniformly bounded by a small constant. However, this assumption is quite strong, as it leads to an optimization problem that depends solely on the noise schedule parameters, ignoring the influence of both the solver and the neural network. While this allows for a fast solution, typically found in a matter of seconds, it overlooks critical information about the pretrained model (trained dataset) and solver design. Furthermore, minimizing the upper bound does not necessarily equate to minimizing the actual global error.

Recent work by Sabour et al. (2024) empirically observe this problem when they derive a bound to the divergence between the analytical ODE solution distribution and the numerical solution distribution. Their objective is challenging to optimize that they need to simulate many sampling trajectories and use a large batch size when performing optimization to reduce the variance and early stopping to prevent divergence. Consequently, their proposed approach is slow and hard to use. Instead of optimizing the global truncation error, Chen et al. (2024) optimizes the local truncation errors. However, their method ignores the information about the solver being used to solve the ODE and it is not guarantee to optimize the global truncation error. Watson et al. (2022; 2021) propose the Differentiable Diffusion Sampler Search (DDSS) method, which aims to improve the Kernel Inception Score by optimizing time discretization. By leveraging Kernel Inception Score (KID) to guide the optimization process, DDSS aims to enhance the quality of generated samples. However, their method requires a large amount of training samples and needs over 50k iterations with batch size 512 to converge. We summarize some key differences between LD3 and similar approaches in Table 1.

## 3 BACKGROUND

Diffusion Probabilistic Models (DPMs) (Ho et al., 2020; Song et al., 2020b) involve a forward diffusion process that gradually converts samples following a data distribution into samples from a pre-specified noise distribution. Specifically, given a sample $\mathbf{x}_0$ from the data distribution $q$, the forward process gradually perturbs it by adding Gaussian noise, which is chosen such that the distribution at time step $T$ is a Gaussian distribution: $\mathbf{x}_T \sim \mathcal{N}(\mathbf{0}, \sigma_T^2 \mathbf{I})$. For any $t \in [0, T]$, the Gaussian transition kernel is defined as

$$q(\mathbf{x}_t|\mathbf{x}_0) = \mathcal{N}(\alpha_t \mathbf{x}_0, \sigma_t^2 \mathbf{I}), \tag{1}$$

where $\alpha_t$ and $\sigma_t$ are two noise schedule hyperparameters designed so that the signal-to-noise ratio (SNR) $\alpha_t^2/\sigma_t^2$ is strictly decreasing when increasing $t$. This ensures that more information about $\mathbf{x}_0$ is discarded as $t$ increases.

To learn the data distribution $q(\mathbf{x}_0)$, DPMs are tasked to recover information discarded by the forward process. This results in a so-called backward process that starts from the noise distribution $\mathbf{x}_T \sim q(\mathbf{x}_T)$ and moves backward through time to reconstruct $\mathbf{x}_0$. Specifically, a neural network $\boldsymbol{\epsilon_\theta}(\mathbf{x}_t, t)$ is trained to predict the noise added by the forward pass given $\mathbf{x}_t$ and $t$ by minimizing

$$\mathbb{E}_{\mathbf{x}_0, \boldsymbol{\epsilon}, t}\left[\omega(t)\|\boldsymbol{\epsilon_\theta}(\mathbf{x}_t, t) - \boldsymbol{\epsilon}\|_2^2\right], \tag{2}$$

where $\mathbf{x}_t := \alpha_t \mathbf{x}_0 + \sigma_t \boldsymbol{\epsilon}$ with $\boldsymbol{\epsilon} \sim \mathcal{N}(\mathbf{0}, \mathbf{I})$ is the noisy sample at time $t$, $t \sim \mathcal{U}[0, T]$ is the time step, and $\omega(t) \in \mathbb{R}^+$ is a time-dependent weight for time step $t$. $\boldsymbol{\epsilon_\theta}(\mathbf{x}_t, t)$ is a learnable deterministic function parameterized by $\boldsymbol{\theta}$ that predicts the noise added to $\mathbf{x}_t$.

One way to sample from a trained DPM is to first draw $\mathbf{x}_T$ randomly and then use it as the initial condition to solve the following *diffusion ODE*, whose solution is proven to match the data distribution $q(\mathbf{x}_0)$ if $\boldsymbol{\epsilon_\theta}(\mathbf{x}_t, t)$ is optimal (Song et al., 2020b; Lu et al., 2022a; Zhang & Chen, 2022; Liu et al., 2022a; Song et al., 2020a; Lu et al., 2022b; Zhao et al., 2023; Zheng et al., 2024):

$$\frac{\partial \mathbf{x}_t}{\partial t} = f(t)\mathbf{x}_t + \frac{g^2(t)}{2\sigma_t}\boldsymbol{\epsilon_\theta}(\mathbf{x}_t, t), \text{ where } f(t) = \frac{\partial \log \alpha_t}{\partial t}, \; g^2(t) = \frac{\partial \sigma_t^2}{\partial t} - 2f(t)\sigma_t^2. \tag{3}$$

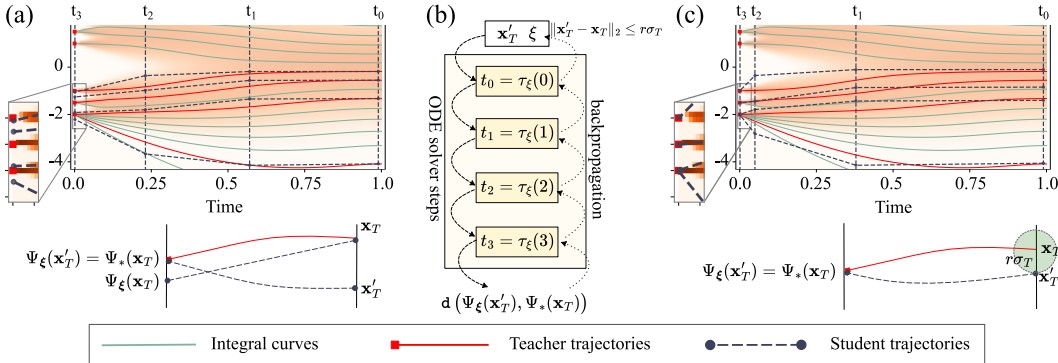

Figure 1: **Motivation and elaboration of LD3.** (a) Directly optimizing the global truncation error loss $\mathcal{L}_{\text{hard}}$ by minimizing the teacher and student outputs improves sample quality. (b) The surrogate objective $\mathcal{L}_{\text{soft}}$ that allows discrepancies in the initial condition (i.e., $\mathbf{x}_T$) between the teacher solver and the student solver is easier to optimize. (c) By optimizing the surrogate objective, LD3 learns better discretization strategies.

## 4 LEARNING TO DISCRETIZE DENOISING DIFFUSION ODES

Our main goal is to reduce the computational burden of DPMs while maintaining their generation quality. In particular, we focus on the small number of neural function evaluation (NFE) regimes with at most 10 evaluations of the denoising network to generate a sample.

The ODE view of DPMs allows us to treat the problem of DPM sampling as solving a class of ODEs, where choosing discretization points is critical. Although one may hope that a universal discretization strategy exists for various DPMs, we find that both the structure of the DPM and the training data influence the optimal time discretization, with no single strategy working well for all cases. Instead, we propose a general yet efficient algorithm to learn discretization strategies for DPMs.

In the following, we first approach this problem under the framework of global truncation error optimization (Sec. 4.1). We then elaborate on a potential underfitting problem of this optimization problem and propose a surrogate objective to improve sampling performance (Sec. 4.2). Finally, we provide details of the algorithm and the training process (Sec. 4.3).

### 4.1 LEARNING TO DISCRETIZE BY OPTIMIZING THE GLOBAL TRUNCATION ERROR

Given a pre-trained denoising network $\boldsymbol{\epsilon}_{\boldsymbol{\theta}}(\cdot, \cdot)$ and an initial state $\mathbf{x}_T \sim \mathcal{N}(\mathbf{0}, \sigma_T^2 \mathbf{I})$ at time step $T$, we solve the diffusion ODE stated in Equation (3), typically by applying a numerical method such as Euler or Heun's method. This process is carried out over a sequence of decreasing time steps $\{t_i\}_{i=0}^N$, where $T = t_0 > t_1 > \cdots > t_N = 0$. At each time step, the state is updated according to the ODE dynamics, guided by the denoising network. The final solution, computed at time $t_N = 0$ is denoted as $\Psi(\mathbf{x}_T, \{t_i\}_{i=0}^N, \boldsymbol{\epsilon}_{\boldsymbol{\theta}})$. Recall that our goal is to learn the set of time steps $\{t_i\}_{i=0}^N$. To ensure these time steps remain monotonic (i.e., non-increasing), we encode the time steps by a monotonic function $\tau_{\boldsymbol{\xi}}$ parameterized by $\boldsymbol{\xi}$ such that for each $i \in \mathbb{N}, 0 \le i \le N$, we have $\tau_{\boldsymbol{\xi}}(i) = t_i$.[1] For simplicity in notation, we now express $\Psi(\mathbf{x}_T, \{\tau_{\boldsymbol{\xi}}(i)\}_{i=0}^N, \boldsymbol{\epsilon}_{\boldsymbol{\theta}})$ as $\Psi_{\boldsymbol{\xi}}(\mathbf{x}_T)$. Additionally, we denote the distribution induced by the transformed state $\Psi_{\boldsymbol{\xi}}(\mathbf{x}_T)$ as $p_{\boldsymbol{\xi}}(\mathbf{x}_0)$, which is obtained by first sampling $\mathbf{x}_T$ and then passing it through the network $\Psi_{\boldsymbol{\xi}}(\cdot)$.

Define $\Psi_*(\mathbf{x}_T, \boldsymbol{\epsilon}_{\boldsymbol{\theta}})$ as a teacher ODE solver that accurately solves the diffusion ODE and the distribution induced by $\Psi_*$ as $q(\mathbf{x}_0)$. We aim to minimize the KL divergence between the teacher distribution $q(\mathbf{x}_0)$ and the student distribution $p_{\boldsymbol{\xi}}(\mathbf{x}_0)$:

$$\min_{\boldsymbol{\xi}} D_{\text{KL}}(q(\mathbf{x}_0) \parallel p_{\boldsymbol{\xi}}(\mathbf{x}_0)) = \min_{\boldsymbol{\xi}} \mathbb{E}_{\mathbf{x}_0 \sim q(\cdot)} \left[ \log \left( \frac{q(\mathbf{x}_0)}{p_{\boldsymbol{\xi}}(\mathbf{x}_0)} \right) \right]. \tag{4}$$

To minimize the objective mentioned above, we focus on reducing the global truncation error, which entails training the student ODE solver $\Psi_{\boldsymbol{\xi}}$ to closely mimic the behavior of the teacher ODE solver $\Psi_*$. We refer to this as *hard* teacher forcing. Formally, this gives rise to the following optimization

---

[1]More details about $\tau_{\boldsymbol{\xi}}$ will be discussed in Section 4.3.

problem we propose:

$$\min_{\boldsymbol{\xi}} \mathcal{L}_{\text{hard}}(\boldsymbol{\xi}) := \min_{\boldsymbol{\xi}} \mathbb{E}_{\mathbf{x}_T \sim \mathcal{N}(\mathbf{0}, \sigma_T^2 \mathbf{I})}[\text{d}(\Psi_{\boldsymbol{\xi}}(\mathbf{x}_T), \Psi_*(\mathbf{x}_T))] \, , \tag{5}$$

where $\text{d}(\cdot, \cdot)$ is a differentiable function that satisfies $\forall \mathbf{x}, \mathbf{y} : \text{d}(\mathbf{x}, \mathbf{y}) \geq 0$ and $\text{d}(\mathbf{x}, \mathbf{y}) = 0$ if and only if $\mathbf{x} = \mathbf{y}$. Some examples are squared $l_2$ distance $\text{d}(\mathbf{x}, \mathbf{y}) = \|\mathbf{x} - \mathbf{y}\|_2^2$ and Learned Perceptual Image Patch Similarity (LPIPS) (Zhang et al., 2018).

For any valid loss in the form of Equation (5), its global optimal solutions are also global optimal solutions to Equation (4). Our proposed objective improves upon previous works by directly optimizing the global truncation error and, consequently, the Kullback-Leibler (KL) divergence between the student and teacher distributions. Unlike existing approaches that focus on minimizing local truncation error (Chen et al., 2024) or optimizing an derived upper bound (Sabour et al., 2024; Xue et al., 2024), our method addresses the fundamental error more effectively. Additionally, our approach leverages information from both the student solver and the trained neural network, which are often overlooked in similar studies (Chen et al., 2024; Xue et al., 2024). This holistic consideration enhances the accuracy and performance of our solution.

### 4.2 Optimizing Discretization Points by Soft Teacher Forcing

Despite having the same global optimum as the KL divergence between the teacher-induced and the student-induced distribution, directly optimizing $\mathcal{L}_{\text{hard}}(\boldsymbol{\xi})$ could lead to underfitting — to minimize the objective, we need to ensure $\Psi_*(\mathbf{x}_T) = \Psi_{\boldsymbol{\xi}}(\mathbf{x}_T)$ for any $\mathbf{x}_T$, which is hard as we are only allowed to optimize $\boldsymbol{\xi}$, which typically contains no more than 20 parameters for student ODE solvers with low NFE. This issue is illustrated by the 1D ODE shown in Figure 1(a), where the green curves are the ground truth integral curves, and the red trajectories are from the teacher. The teacher matches the ground truth closely, as it can take fine-grained steps to solve the ODE. However, given the restriction that the student solver can only evaluate the ODE at three time steps before generating the output, an inevitable truncation error exists between the teacher and the student.

One way around this problem is to optimize the parameters of the student denoising network $\epsilon_{\boldsymbol{\theta}}$ in addition to $\boldsymbol{\xi}$. However, this will significantly increase the sample complexity and the training time, limiting the method's efficiency and portability. Instead, we propose to relax the "hard" teacher forcing criterion. Specifically, for any $\mathbf{x}_T$ and the corresponding output of the teacher $\mathbf{x}_0 := \Psi_*(\mathbf{x}_T)$, we only require the existence of an input $\mathbf{x}_T'$ that is "close" to $\mathbf{x}_T$, such that the student's output given $\mathbf{x}_T'$ (i.e., $\Psi_{\boldsymbol{\xi}}(\mathbf{x}_T')$) matches $\mathbf{x}_0$. We define $B(\mathbf{x}, r\sigma_T) := \{\mathbf{x}' \mid \|\mathbf{x} - \mathbf{x}'\|_2 \leq r\sigma_T\}$ as the L2 ball of radius $r\sigma_T$ around $\mathbf{x}$. Take the 1D ODE in Figure 1(a) as an example. Although it may be impossible to force the student solvers to map the same point $\mathbf{x}_T$ to the teacher's output, we can instead find a nearby input $\mathbf{x}_T'$ that does map to the teacher's output $\Psi_*(\mathbf{x}_T)$ (i.e., $\Psi_{\boldsymbol{\xi}}(\mathbf{x}_T')$). The goal is that, by finding such $\mathbf{x}_T'$ points close to $\mathbf{x}_T$, as illustrated in Figure 1(c), the student solver's distribution at $t = 0$ will still closely match the teacher's distribution. Note that $\mathbf{x}_T'$ is only used during training to find the optimal discretization points. Formally, we relax the objective defined in Equation (5) into the following for an $r > 0$:

$$\min_{\boldsymbol{\xi}} \mathcal{L}_{\text{soft}}(\boldsymbol{\xi}) := \min_{\boldsymbol{\xi}} \mathbb{E}_{\mathbf{x}_T \sim \mathcal{N}(\mathbf{0}, \sigma_T^2 \mathbf{I})} \left[ \min_{\mathbf{x}_T' \in B(\mathbf{x}_T, r\sigma_T)} \text{d}(\Psi_{\boldsymbol{\xi}}(\mathbf{x}_T'), \Psi_*(\mathbf{x}_T)) \right]. \tag{6}$$

The effectiveness of this relaxed objective depends on two questions: (i) compared to $\mathcal{L}_{\text{hard}}(\boldsymbol{\xi})$, how much easier is it to optimize $\mathcal{L}_{\text{soft}}(\boldsymbol{\xi})$ given the fact that we only have a handful of learnable parameters (i.e., $\boldsymbol{\xi}$); (ii) whether minimizing $\mathcal{L}_{\text{soft}}(\boldsymbol{\xi})$ leads to student distributions that have small KL divergence with the teacher distribution (i.e., the ground-truth objective in Eq. (4)).

We start by showing positive evidence to the first question — empirically, a small $r$ suffices to ensure $\mathcal{L}_{\text{soft}}(\boldsymbol{\xi})$ being much smaller than $\mathcal{L}_{\text{hard}}(\boldsymbol{\xi})$ after training. We experiment on a pre-trained DPM (Karras et al., 2022) on AFHQv2 (Choi et al., 2020). We optimize the discretization points (i.e., $\boldsymbol{\xi}$) with respect to $\mathcal{L}_{\text{soft}}(\boldsymbol{\xi})$

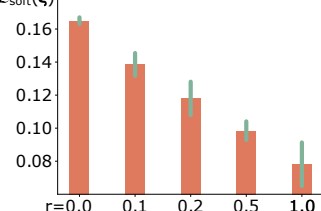

Figure 2: $\mathcal{L}_{\text{soft}}(\boldsymbol{\xi})$ drops significantly as we increase $r$.

using different $r$ and plot the training loss. We use the LPIPS distance (Zhang et al., 2018) for $\text{d}(\cdot, \cdot)$. As shown in Figure 2, compared to $r = 0.0$, where $\mathcal{L}_{\text{soft}}$ and $\mathcal{L}_{\text{hard}}$ are identical, the loss is significantly reduced as we relax the optimization problem by increasing $r$.

We now move on to the second question: if we can effectively minimize $\mathcal{L}_{\text{soft}}(\boldsymbol{\xi})$, can we establish some form of guarantee of the student in terms of its KL divergence with the teacher distribution? We confirm this with the following theoretical result.

---

**Theorem 1.** *Let $\Psi_*$ and $\Psi_{\boldsymbol{\xi}}$ be a teacher and student ODE solver each with noise distribution $\mathcal{N}(\mathbf{0}, \sigma_T^2\mathbf{I}) \in \mathbb{R}^d$, and with, respectively, distributions $q$ and $p_{\boldsymbol{\xi}}$. Assume both $\Psi_*$ and $\Psi_{\boldsymbol{\xi}}$ are invertible. Let $r > 0$, if the objective from Equation* (6) *has an optimal solution $\boldsymbol{\xi}^*$ for $r$ with objective value* 0, *we have*

$$D_{\text{KL}}(q(\mathbf{x}) \,\|\, p_{\boldsymbol{\xi}^*}(\mathbf{x})) \le \frac{r^2}{2} + r\sqrt{d+1} + \mathbb{E}_{\mathbf{x} \sim q(\mathbf{x})}\left[|C(\Psi_*(\mathbf{x})) - C(\Psi_{\boldsymbol{\xi}^*}(\mathbf{x}))|\right],$$

*where* $C(\Psi_{\boldsymbol{\xi}^*}(\mathbf{x})) = \log|\det J_{\Psi_{\boldsymbol{\xi}^*}}(\Psi_{\boldsymbol{\xi}^*}^{-1}(\mathbf{x}))|.$

---

The proof is provided in Appendix A.1. Intuitively, the theorem states that, if we can find an optimal solution $\boldsymbol{\xi}^*$ that minimizes $\mathcal{L}_{\text{soft}}$, then the KL divergence between the teacher (i.e., $q(\mathbf{x})$) and the student (i.e., $p_{\boldsymbol{\xi}^*}(\mathbf{x})$) can be upper bounded. The first two terms depend mainly on $r$ and the square root of the dimensionality: $\sqrt{d+1}$. Since $r$ is chosen to be quite small (e.g., 0.19 for CIFAR10, 4 NFE), the first two terms are effectively tight in practice. While it is hard to establish an analytic bound for the third term, we conduct numerical evaluations to estimate its magnitude in practice and observe that it reduces with $r$. See Appendix A.2 for more details.

### 4.3 PRACTICAL IMPLEMENTATION

---

**Algorithm 1** LD3

---

**Require:** Student solver $\Psi_{\boldsymbol{\xi}}$, teacher solver $\Psi_*$, and $r$
1: $\mathcal{D} \leftarrow \{(\mathbf{x}_T', \mathbf{x}_T, \Psi_*(\mathbf{x}_T)) \,|\, \mathbf{x}_T \sim \mathcal{N}(\mathbf{0}, \sigma_T^2), \mathbf{x}_T' = \mathbf{x}_T\}$  $\triangleright$ Generate data $\mathcal{D}$
2: **while** not converged **do**
3: $\quad (\mathbf{x}_T', \mathbf{x}_T, \Psi_*(\mathbf{x}_T)) \sim \mathcal{D}$
4: $\quad \mathcal{L}(\boldsymbol{\xi}, \boldsymbol{\xi}^c, \mathbf{x}_T') = \text{LPIPS}(\Psi_{\boldsymbol{\xi}, \boldsymbol{\xi}^c}(\mathbf{x}_T'), \Psi_*(\mathbf{x}_T))$ subject to $\mathbf{x}_T' \in B(\mathbf{x}_T, r\sigma_T)$
5: $\quad$ Update $\boldsymbol{\xi}, \boldsymbol{\xi}^c$, and $\mathbf{x}_T'$ using the corresponding gradients $\nabla \mathcal{L}(\boldsymbol{\xi}, \boldsymbol{\xi}^c, \mathbf{x}_T')$
6: $\quad \mathbf{x}_T' \leftarrow \mathbf{x}_T + \mathbb{1}\left[\|\mathbf{x}_T' - \mathbf{x}_T\|_2 > r\right] \cdot r \frac{\mathbf{x}_T' - \mathbf{x}_T}{\|\mathbf{x}_T' - \mathbf{x}_T\|_2}$  $\triangleright$ Projected SGD
7: $\quad$ Update $\mathcal{D}$ with the new $\mathbf{x}_T'$
8: **end while**

---

Now that we have justified the effectiveness of $\mathcal{L}_{\text{soft}}(\boldsymbol{\xi})$, we are left with the question of how to optimize it in practice. This is achieved by treating $\mathcal{L}_{\text{soft}}(\boldsymbol{\xi})$ as jointly optimizing $\boldsymbol{\xi}$ and $\mathbf{x}_T'$ with the constraint that $\mathbf{x}_T'$ is within the $r$-ball of $\mathbf{x}_T$:

$$\mathcal{L}(\boldsymbol{\xi}, \mathbf{x}_T') := \mathbb{E}_{\mathbf{x}_T \sim \mathcal{N}(\mathbf{0}, \sigma_T^2\mathbf{I})}[\text{LPIPS}(\Psi_{\boldsymbol{\xi}}(\mathbf{x}_T'), \Psi_*(\mathbf{x}_T))], \quad \text{subject to } \mathbf{x}_T' \in B(\mathbf{x}_T, r\sigma_T). \quad (7)$$

Note that we choose d := LPIPS (Zhang et al., 2018) as the distance metric in our setting. LPIPS is also a common choice in many distillation-based models (Song et al., 2023; Salimans & Ho, 2022). As illustrated in Figure 1(b), given an input-output pair $(\mathbf{x}_T, \Psi_*(\mathbf{x}_T))$ from the teacher, we forward through the student ODE solver steps with learnable parameters $\boldsymbol{\xi}$ and $\mathbf{x}_T'$. Backpropagation is then performed to get the gradients w.r.t. $\text{LPIPS}(\Psi_{\boldsymbol{\xi}}(\mathbf{x}_T'), \Psi_*(\mathbf{x}_T))$. Finally, we use projected SGD to enforce the constraint on $\mathbf{x}_T'$ and use SGD to update $\boldsymbol{\xi}$. Specifically, let $\mathbf{x}_T$ be the center of a $r$-sphere and $\mathbf{x}_T'$ a point. Then the projection of $\mathbf{x}_T'$, which is the intersection of the line between $\mathbf{x}_T$ and $\mathbf{x}_T'$ with the sphere's surface, can be computed as $\mathbf{x}_p := \mathbf{x}_T + r\frac{\mathbf{x}_T' - \mathbf{x}_T}{\|\mathbf{x}_T' - \mathbf{x}_T\|_2}$.

**Parameterization.** Starting from a trainable vector $\boldsymbol{\xi} \in \mathbb{R}^{N+1}$, we model $\tau_{\boldsymbol{\xi}}(i)$ as a strictly monotonically decreasing function using a cumulative softmax function, followed by renormalization to the range $[t_{\min}, T]$:

$$\tau_{\boldsymbol{\xi}}(i) := \frac{\tau_{\boldsymbol{\xi}}'(i) - \tau_{\boldsymbol{\xi}\,\min}'}{\tau_{\boldsymbol{\xi}\,\max}' - \tau_{\boldsymbol{\xi}\,\min}'}(T - t_{\min}) + t_{\min}, \quad \text{where} \quad \tau_{\boldsymbol{\xi}}'(i) = \sum_{n=i}^{N} \text{softmax}(\boldsymbol{\xi})[n].$$

Here, $t_{\min}$ is often utilized in training and sampling as a substitute for 0 in the diffusion model to mitigate numerical instability problems (Karras et al., 2022; Song et al., 2023).

During training, DPMs denoise images from real data, while during inference, they use predictions from previous steps, leading to discrepancies and errors (Ning et al., 2023; Li et al., 2023). As suggested by (Li et al., 2023), we learn decoupled time steps $t_i^c$ as input to the denoising model, while $t_i$ determines the solver's step size. We parameterize $t_i^c$ as $\tau_{\boldsymbol{\xi}}^c(i) := \tau_{\boldsymbol{\xi}}(i) + \boldsymbol{\xi}_i^c$. For simplicity, we still use $\Psi_{\boldsymbol{\xi}}$ to refer to the student solver with decoupled time step variables.

**Training.** The final LD3 algorithm is shown in Algorithm 1. In line 1, we first generate training samples by first sample $\mathbf{x}_T \sim \mathcal{N}(\mathbf{0}, \sigma_T \mathbf{I})$ and then compute the corresponding teacher outputs $\Psi_*(\mathbf{x}_T)$. Initially, the same starting samples are used for both the student and the teacher (i.e., $\mathbf{x}_T' = \mathbf{x}_T$), resulting in a dataset $\mathcal{D} = \{(\mathbf{x}_T', \mathbf{x}_T, \Psi_*(\mathbf{x}_T))\}$. In every iteration, we first compute the objective shown in Equation (7) (line 4), and then apply gradient-based updates to $\boldsymbol{\xi}$ and $\mathbf{x}_T'$ (line 5). An additional projected SGD step is applied to $\mathbf{x}_T'$ to bound the distance between $\mathbf{x}_T'$ and $\mathbf{x}_T$ (line 6). The algorithm terminates after convergence.

Since computing $\Psi_{\boldsymbol{\xi}}(\mathbf{x}_T')$ involves evoking the denoising network $\boldsymbol{\epsilon}_{\boldsymbol{\theta}}$ multiple times, naively storing all intermediate outputs for efficient backpropagation would lead to memory overhead that scales linearly w.r.t. NFE. To reduce the memory overhead, we use the rematerialization technique proposed in (Watson et al., 2022) to only cache the intermediate $\mathbf{x}_t$. This leads to an almost constant memory overhead w.r.t. NFE. Please refer to Appendix D.3 for details.

## 5 EXPERIMENTS

**Experiment setup.** We evaluate 7 pre-trained diffusion models across different domains. For pixel space models, we include CIFAR10 (32×32) (Krizhevsky & Hinton, 2009), FFHQ (64×64) (Karras et al., 2019), and AFHQv2 (64×64) (Choi et al., 2020). For latent space models, we assess LSUN-Bedroom (256×256) (Yu et al., 2015) and class-conditional ImageNet (256×256) (Russakovsky et al., 2015) with a guidance scale of 2.0. Additionally, we consider text-to-image generation models, including Stable Diffusion v1.5 (Rombach et al., 2022) at 512×512 pixels with a guidance scale of 7.5, and InstaFlow Liu et al. (2023).

We primarily assess LD3 using advanced diffusion ODE solvers, including DPM_solver++ (Lu et al., 2022b), Uni_PC (Zhao et al., 2023), and iPNDM (Zhang & Chen, 2022). Additionally, we test LD3 with Euler, a standard black-box ODE solver. To evaluate the performance of our learned discretization method, we compare it against 8 existing discretization methods. For commonly used discretization choices, we include *time uniform* (Lu et al., 2022b; Ho et al., 2020), *time quadratic* (Song et al., 2020a), *time EDM* (Karras et al., 2022), and *time logSNR* (uniform in $\lambda$) (Zhang & Chen, 2022), with details provided in Appendix D.2. For more recent advanced discretization methods, we compare LD3 to DMN (Xue et al., 2024), GITS (Chen et al., 2024), AYS (Sabour et al., 2024), and Watson et al. (2022). Since AYS (Sabour et al., 2024) and Watson et al. (2022) do not have published code, we adhere to their settings and compare our results with their reported metrics in Appendix E.

For CIFAR10, FFHQ, and AFHQv2, we use 100 samples for both training and validation and train LD3 for 7 epochs with a batch size of 2. We set $r$ proportional to the dimensionality $d$ and inversely proportional to the squared NFE: $r = \gamma \times \frac{d}{\text{NFE}^2}$, where $\gamma = 0.001$ in all experiments. For Latent Diffusion (Rombach et al., 2022) on ImageNet and LSUN-Bedroom, we use 100 samples for both training and validation, with the training conducted over 5 epochs. Unless stated otherwise, we draw 50k samples for the evaluation using the FID score (Heusel et al., 2017) against a reference data set, where lower scores indicate better quality. For text-to-image generation, we train Stable Diffusion (Rombach et al., 2022) and InstaFlow Liu et al. (2023) by randomly selecting 5 prompts from the MSCOCO dataset (Lin et al., 2015) and generate 10 training pairs for each prompt. We train them for 5 epochs. Details of experiment settings can be found in Appendix D.

| LD3 | DMN | GITS | | LD3 | DMN | GITS |

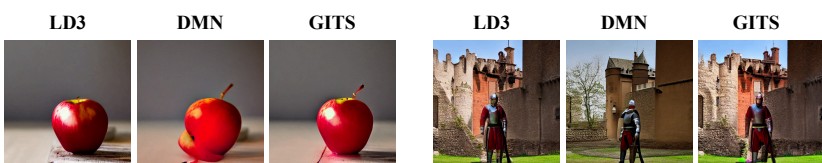

"A red apple on a white table."        "A medieval knight standing in a castle courtyard."

Figure 3: Side-by-side comparison of selected images generated with Stable Diffusion [iPNDM]. Left: NFE=6, Right: NFE=5.

**Main results.** LD3 consistently improves generation quality across all solvers (Table 2), particularly in low NFE settings. For instance, on the AFHQv2 dataset with NFE=4, iPNDM [LD3] achieves an FID score of **9.96**, outperforming other approaches, which achieve a best score of 12.89. At NFE=10, LD3 delivers the best performance across all datasets, with FID scores of **2.38** on CIFAR10, **2.27** on AFHQv2, and **3.25** on FFHQ. Additional results, including for the pixel space LSUN-Bedroom (256x256) dataset and various NFE settings, are available in Appendix F.1 and Appendix F.4. Table 3 compares LD3's optimized time steps with common time discretization methods, demonstrating a clear advantage of the optimized steps over standard choices.

Table 2: FID comparison on CIFAR10, AFHQv2, and FFHQ and the two solvers Uni_PC and iP-NDM. We compare LD3 and two different time discretization optimization methods, DMN and GITS. FID scores are computed with 50k samples using the reference data set.

| Method | NFE=4 | NFE=6 | NFE=8 | NFE=10 |
|---|---|---|---|---|
| **CIFAR10** | | | | |
| Uni_PC (3M) | 43.92 | 13.12 | 4.41 | 3.16 |
| Uni_PC [GITS] | 25.32 | 11.19 | 5.67 | 3.70 |
| Uni_PC [DMN] | 26.35 | 8.09 | 5.90 | **2.45** |
| Uni_PC [LD3] | 13.72 | 5.92 | 3.42 | 2.87 |
| iPNDM (3M) | 35.04 | 11.80 | 5.67 | 3.69 |
| iPNDM [GITS] | 15.63 | 6.82 | 4.29 | 2.78 |
| iPNDM [DMN] | 28.09 | 9.24 | 7.68 | 3.31 |
| iPNDM [LD3] | **9.31** | **3.35** | **2.81** | **2.38** |
| Teacher | | | 2.08 | |
| **AFHQv2** | | | | |
| Uni_PC (3M) | 33.78 | 8.27 | 4.60 | 3.81 |
| Uni_PC [GITS] | **12.20** | 7.26 | 3.86 | 2.88 |
| Uni_PC [DMN] | 30.32 | 14.46 | 6.85 | 2.94 |
| Uni_PC [LD3] | 12.99 | **3.81** | **2.90** | **2.84** |
| iPNDM (3M) | 23.20 | 9.55 | 4.49 | 3.19 |
| iPNDM [GITS] | 12.89 | 6.10 | 4.03 | 3.26 |
| iPNDM [DMN] | 33.15 | 16.01 | 10.12 | 3.22 |
| iPNDM [LD3] | **9.96** | **3.63** | **2.63** | **2.27** |
| Teacher | | | 2.11 | |
| **FFHQ** | | | | |
| Uni_PC (3M) | 53.25 | 11.24 | 5.59 | 3.90 |
| Uni_PC [GITS] | 21.38 | 12.21 | 7.84 | 4.46 |
| Uni_PC [DMN] | 25.82 | 9.47 | 6.85 | 3.54 |
| Uni_PC [LD3] | **21.00** | **5.97** | **3.50** | **3.27** |
| iPNDM (3M) | 36.54 | 16.44 | 8.11 | 5.39 |
| iPNDM [GITS] | 18.05 | 9.38 | 5.72 | 3.96 |
| iPNDM [DMN] | 31.30 | 12.12 | 11.00 | 5.24 |
| iPNDM [LD3] | **17.96** | **6.47** | **3.97** | **3.25** |
| Teacher | | | 2.54 | |

Table 4: FID score at small NFE regimes on latent diffusion models. We investigate LSUN-Bedroom and ImageNet datasets. FID scores are computed based on 50k samples using the reference data set. Additional results can be found in Appendix F.4.

| Method | NFE=4 | NFE=5 | NFE=6 | NFE=7 |
|---|---|---|---|---|
| **LSUN-Bedroom-256 (latent space)** | | | | |
| Uni_PC (3M) | 39.78 | 13.88 | 6.57 | 4.56 |
| Uni_PC [GITS] | 70.93 | 47.37 | 22.33 | 17.27 |
| Uni_PC [DMN] | 29.22 | **8.21** | **4.40** | 4.55 |
| Uni_PC [LD3] | **20.15** | 9.09 | 4.98 | **4.18** |
| iPNDM (3M) | 11.93 | 6.38 | 5.08 | 4.39 |
| iPNDM [GITS] | 76.86 | 59.17 | 28.09 | 19.54 |
| iPNDM [DMN] | 11.82 | 6.15 | 4.71 | 5.16 |
| iPNDM [LD3] | **8.48** | **5.93** | **4.52** | **4.31** |
| Teacher | | 3.06 | | |
| **Imagenet-256 (latent space)** | | | | |
| Uni_PC (3M) | 20.01 | 8.51 | 5.92 | 5.20 |
| Uni_PC [GITS] | 54.88 | 34.91 | 14.62 | 9.04 |
| Uni_PC [DMN] | 16.72 | 7.96 | 7.54 | 7.81 |
| Uni_PC [LD3] | **9.89** | **5.03** | **4.46** | **4.32** |
| iPNDM (3M) | 13.86 | 7.80 | 6.03 | 5.35 |
| iPNDM [GITS] | 56.00 | 43.56 | 19.33 | 10.33 |
| iPNDM [DMN] | 10.15 | 7.33 | 7.25 | 7.40 |
| iPNDM [LD3] | **9.19** | **6.03** | **5.09** | **4.68** |
| Teacher | | 4.17 | | |

Table 5: FID scores on Stable Diffusion v1.5. We follow the standard FID evaluation with 30k captions from MS-COCO (Lin et al., 2015).

| Method | NFE=4 | NFE=5 | NFE=6 | NFE=7 |
|---|---|---|---|---|
| iPNDM (2M) | 17.76 | 14.41 | 13.86 | 13.76 |
| iPNDM [GITS] | 18.05 | 14.11 | **12.10** | **11.80** |
| iPNDM [DMN] | 21.70 | 17.30 | 13.68 | 11.88 |
| iPNDM [LD3] | **17.32** | **13.07** | 12.40 | 11.83 |

Table 3: Comparison of FID scores on CIFAR10 using iPNDM solver. We compare LD3's discretization with commonly selected heuristics.

| Discretization type | NFE=4 | NFE=6 | NFE=8 | NFE=10 |
|---|---|---|---|---|
| Time LogSNR | 35.04 | 11.80 | 5.67 | 3.69 |
| Time Uniform | 266.26 | 229.39 | 205.24 | 185.28 |
| Time Quadratic | 139.72 | 68.82 | 37.82 | 23.40 |
| Time EDM | 29.78 | 9.95 | 5.41 | 3.80 |
| LD3 | **9.31** | **3.35** | **2.81** | **2.38** |

Table 6: LD3 improves InstaFlow, a few-step text-to-image generation model. We compare the FID score with 10k captions sampled from MS-COCO (Lin et al., 2015).

| Method | NFE=2 | NFE=4 | NFE=6 |
|---|---|---|---|
| InstaFlow | 22.56 | 16.04 | 14.78 |
| InstaFlow [LD3] | **15.49** | **14.33** | **14.12** |
| Teacher (NFE = 8, Uniform) | | 14.25 | |

For latent space diffusion models, we compare our method to the best discretization approaches using Latent Diffusion models (Rombach et al., 2022) trained on LSUN-Bedroom and ImageNet, as shown in Table 4. Our model consistently matches or outperforms baseline methods, particularly

in low NFE scenarios. For example, at NFE=4, Uni_PC [LD3] achieves an FID score of **20.15** on LSUN-Bedroom, around 9 points better than the next best model, Uni_PC [DMN].

We also test the performance of LD3 on text-to-image generation. Table 5 shows performance of LD3 on Stable Diffusion. Generally, LD3 outperforms the default Time Uniform discretization while GITS and DMN only improve the performance given enough number of steps. InstaFlow (Liu et al., 2023) is a few-step text-to-image generation, it can generate high-quality images in just a few steps using a simple Euler solver. We further show that LD3 can significantly boost the generation quality by optimizing the time step (Table 6). For example, LD3 improves FID score from 22.56 to **15.49** with only 2 NFE. One might question how different training prompts affect the final results. Interestingly, when we train our model using different sets of prompts, we achieve similar FID scores with small variance. Please refer to Appendix F.3 for more details. For qualitative comparisons, please refer to Figures 3 and 4 and Appendix F.5.

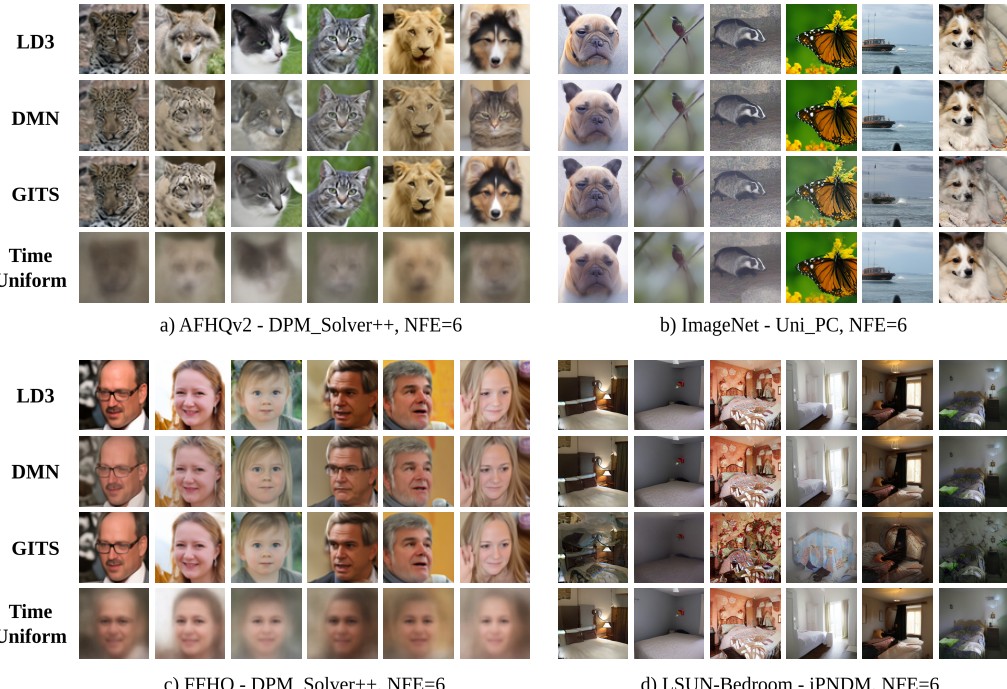

a) AFHQv2 - DPM_Solver++, NFE=6

b) ImageNet - Uni_PC, NFE=6

c) FFHQ - DPM_Solver++, NFE=6

d) LSUN-Bedroom - iPNDM, NFE=6

Figure 4: Side-by-side comparison of random images generated by different pre-trained models across four datasets: AFHQv2, ImageNet, FFHQ, and LSUM-Bedroom. We compare LD3 with DMN (Xue et al., 2024), GITS (Chen et al., 2024), and Time Uniform discretization. For each dataset, samples from each column are created using the same initial noise, solver, and the number of NFE. We provide more side-by-side comparisons in Appendix F.5

**Ablation study.** We investigate the importance of different components in our model as shown in Table 7. Initially, we compare variants with and without training $\xi^c$ (cf. the decoupling technique described in Section 4.3) and then examine the effects of other factors on performance. This is particularly important because the variant without $\xi^c$ is easily integrated into any ODE samplers, making a detailed study of its behavior beneficial to the community. We observe that $\xi^c$ significantly contributes to the performance of LD3, which is expected as it effectively doubles the number of trainable parameters. For both versions, optimizing $\mathcal{L}_{\text{hard}}$ results in worse performance compared to optimizing $\mathcal{L}_{\text{soft}}$. For instance, on CIFAR10, with NFE=6, using $\mathcal{L}_{\text{soft}}$ improves the FID score by $44\%$ (from 13.46 to 7.51). The effect of $\mathcal{L}_{\text{soft}}$ is less pronounced without $\xi^c$; with the same NFE, $\mathcal{L}_{\text{soft}}$ boosts the FID by $15\%$ (from 14.20 to 12.03).

Table 8 highlights the importance of learning a time discretization tailored to the ODE solver type. In particular, we train LD3 using DPM_Solver++, utilizing the optimized time steps for testing on both DPM_Solver++ and Euler. If the choice of solver were irrelevant, the optimized time steps would perform well on both solvers. However, this is not the case. Optimized time steps for DPM_Solver++ perform poorly for Euler and vice versa. For instance, the optimized time steps for DPM_Solver++

yield an FID score of 42.44 on Euler, whereas those optimized for Euler achieve a score of 25.28 on the same solver.

Table 7: Ablation study on CIFAR10 and FFHQ. We conduct the experiment with DPM_Solver++(3M).

|  |  | Setting | NFE=4 | NFE=5 | NFE=6 |
|---|---|---|---|---|---|
| **CIFAR10** | w/ $\xi^c$ | Full setting | **19.39** | **9.08** | **7.51** |
|  |  | w/ $\mathcal{L}_{hard}$ | 19.62 | 10.17 | 13.46 |
|  |  | w/ L2 | 33.25 | 15.13 | 11.78 |
|  | w/o $\xi^c$ | Full setting | 33.34 | 19.39 | 12.03 |
|  |  | w/ $\mathcal{L}_{hard}$ | 33.75 | 19.50 | 14.20 |
|  |  | w/ L2 | 56.49 | 21.65 | 15.36 |
| **FFHQ** | w/ $\xi^c$ | Full setting | **27.99** | **13.32** | **7.53** |
|  |  | w/ $\mathcal{L}_{hard}$ | 30.52 | 14.20 | 9.12 |
|  |  | w/ L2 | 26.01 | 16.33 | 13.47 |
|  | w/o $\xi^c$ | Full setting | 30.94 | 16.76 | 11.05 |
|  |  | w/ $\mathcal{L}_{hard}$ | 31.04 | 17.64 | 12.58 |
|  |  | w/ L2 | 38.73 | 21.97 | 16.57 |

Table 8: We examine the importance of tailoring the time discretization to ODE solvers. We use Euler and DPM_Solver++ with LD3 on the AFHQv2 dataset for this experiment. Each row shows the FID scores when LD3 is trained with a specific solver, while each column presents the FID scores when sampled with a specific solver.

|  | Solver name | DPM_Solver++ | Euler_Solver |
|---|---|---|---|
| NFE=4 | DPM_Solver++ | 13.86 | 42.44 |
|  | Euler_Solver | 231.00 | 25.28 |
| NFE=6 | DPM_Solver++ | 5.47 | 14.16 |
|  | Euler_Solver | 46.63 | 11.08 |

**Efficiency and performance.** LD3 demonstrates significantly faster training times compared to AYS (Sabour et al., 2024). As shown in Figure 5 (a), LD3 can be optimized in less than an hour on a single GPU, while AYS typically requires several hours on multiple GPUs. For instance, at 10 NFE, our model needs approximately 36 minutes on a single NVIDIA A100 GPU, whereas AYS requires 3 to 4 hours on 8 NVIDIA RTX6000s.

We generally only need 100 teacher noise-sample pairs for training and validation. Figure 5 (b) demonstrates that increasing the training size reduces the FID score. Since the performance plateaus at around 100 samples, we use ∼100 samples in practice to balance performance and efficiency.

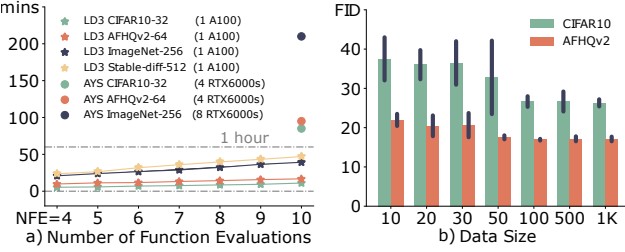

Figure 5: (a) Training time for LD3 and AYS (Sabour et al., 2024) on various NFE; (b) The effect of data size on model performance (FID is evaluated with 5K samples, DPM_Solver++(3M), NFE=4).

Figure 6: Comparison of global truncation error across time discretization methods. RMSD is computed between student and goal samples using 100-step DDIM.

## 6    CONCLUSION AND LIMITATIONS

We introduced LD3, a lightweight framework designed to reduce computational costs when sampling from pre-trained DPMs. LD3 learns time discretization for ODE sampling, significantly lowering the NFE needed to generate high-quality images with minimal training overhead. Our experiments across various datasets demonstrate that LD3 consistently improves sampling quality. For instance, LD3 reduced the FID score on CIFAR10 (4 NFE) from 35.04 to 9.31 with 5 minutes of training on a GPU. These results indicate that LD3 provides a more efficient approach to sampling from pre-trained diffusion models, with promising applications in image synthesis and beyond.

**Limitations and broader impact.** Despite significant performance improvements for a few NFE, LD3 still falls short of distillation-based methods regarding sample quality. A limitation of LD3 is that it needs to be trained separately for each given number of NFEs. Additionally, our model necessitates a differentiable solver, which may not always be feasible. Using LD3 with advanced diffusion ODE solvers to generate fake content could also exacerbate the potential risks of LD3 being used for malicious purposes.

ACKNOWLEDGEMENTS

This work was funded in part by Deutsche Forschungsgemeinschaft (DFG, German Research Foundation) under Germany's Excellence Strategy - EXC 2075 – 390740016, the DARPA ANSR program under award FA8750-23-2-0004, the DARPA CODORD program under award HR00112590089, the NSF grant #IIS-1943641, and gifts from Adobe Research, Cisco Research, and Amazon. We acknowledge the support of the Stuttgart Center for Simulation Science (SimTech). VT and MN thank IMPRS-IS (International Max Planck Research School for Intelligent Systems) for the support. This work also received partial support from the Diabetes Center Berne.

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

# LEARNING TO DISCRETIZE DIFFUSION ODES
## ADDITIONAL MATERIAL

# A ANALYTICAL PROOFS

## A.1 PROOF OF THEOREM

> **Theorem 1.** *Let $\Psi_*$ and $\Psi_{\boldsymbol{\xi}}$ be a teacher and student ODE solver each with noise distribution $\mathcal{N}(\mathbf{0}, \sigma_T^2 \mathbf{I})$, $\mathbf{0} \in \mathbb{R}^d$, and with, respectively, distributions $q$ and $p_{\boldsymbol{\xi}}$. Assume both $\Psi_*$ and $\Psi_{\boldsymbol{\xi}}$ are invertible. Let $r > 0$, if the objective from Equation (6) has an optimal solution $\boldsymbol{\xi}^*$ for $r$ with objective value $0$, we have*
>
> $$D_{\mathrm{KL}}(q(\mathbf{x}) \parallel p_{\boldsymbol{\xi}^*}(\mathbf{x})) \leq \frac{r^2}{2} + r\sqrt{d+1} + \mathbb{E}_{\mathbf{x} \sim q(\mathbf{x})}\left[|C(\Psi_*(\mathbf{x})) - C(\Psi_{\boldsymbol{\xi}^*}(\mathbf{x}))|\right],$$
>
> *where $C(\Psi_{\boldsymbol{\xi}^*}(\mathbf{x})) = \log|\det J_{\Psi_{\boldsymbol{\xi}^*}}(\Psi_{\boldsymbol{\xi}^*}^{-1}(\mathbf{x}))|$.*

*Proof.* By assuming the invertibility of the solvers and the loss of Equation (6) having an optimal (zero loss and satisfying all $r\sigma_T$-ball constraints) solution $\boldsymbol{\xi}^*$, we have for every $\mathbf{x} \sim q(\mathbf{x})$ exactly one $\boldsymbol{b}$ with $\Psi_*^{-1}(\mathbf{x}) = \boldsymbol{b}$ and exactly one corresponding $\boldsymbol{a}$ with $\Psi_{\boldsymbol{\xi}^*}^{-1}(\mathbf{x}) = \boldsymbol{a}$. Moreover, by definition of the loss objective and the fact that $\boldsymbol{a}$ is an optimal and, therefore, also feasible solution, we have $\boldsymbol{a} \in B(\boldsymbol{b}, r\sigma_T)$ and thus $\|\boldsymbol{a} - \boldsymbol{b}\|_2 \leq r\sigma_T$.

Using the density function of the normal distribution, we can now write

$$\mathbb{E}_{\mathbf{x} \sim q(\mathbf{x})}\left[\log\left(\frac{q(\mathbf{x})}{p_{\boldsymbol{\xi}}(\mathbf{x})}\right)\right]$$

$$= \mathbb{E}_{\mathbf{x} \sim q(\mathbf{x})}\left[\log\left(\frac{\mathcal{N}(\boldsymbol{b})\left|\det \frac{d\Psi_*(\boldsymbol{b})}{d\boldsymbol{b}}\right|^{-1}}{\mathcal{N}(\boldsymbol{a})\left|\det \frac{d\Psi_{\boldsymbol{\xi}^*}(\boldsymbol{a})}{d\boldsymbol{a}}\right|^{-1}}\right)\right]$$

$$= \mathbb{E}_{\mathbf{x} \sim q(\mathbf{x})}\left[\log(\mathcal{N}(\boldsymbol{b})) + \log\left(\left|\det \frac{d\Psi_*(\boldsymbol{b})}{d\boldsymbol{b}}\right|^{-1}\right) - \log(\mathcal{N}(\boldsymbol{a})) - \log\left(\left|\det \frac{d\Psi_{\boldsymbol{\xi}^*}(\boldsymbol{a})}{d\boldsymbol{a}}\right|^{-1}\right)\right]$$

$$= \mathbb{E}_{\mathbf{x} \sim q(\mathbf{x})}\left[\log(\mathcal{N}(\boldsymbol{b})) - \log(\mathcal{N}(\boldsymbol{a}))\right] + \mathbb{E}_{\mathbf{x} \sim q(\mathbf{x})}\left[\log\left(\left|\det \frac{d\Psi_*(\boldsymbol{b})}{d\boldsymbol{b}}\right|^{-1}\right) - \log\left(\left|\det \frac{d\Psi_{\boldsymbol{\xi}^*}(\boldsymbol{a})}{d\boldsymbol{a}}\right|^{-1}\right)\right]$$

$$= \mathbb{E}_{\mathbf{x} \sim q(\mathbf{x})}\left[\log\left(\frac{\prod_{i=1}^d \frac{1}{\sigma_T\sqrt{2\pi}}\exp\left(-\frac{1}{2}\frac{b_i^2}{\sigma_T^2}\right)}{\prod_{i=1}^d \frac{1}{\sigma_T\sqrt{2\pi}}\exp\left(-\frac{1}{2}\frac{a_i^2}{\sigma_T^2}\right)}\right)\right] + \mathbb{E}_{\mathbf{x} \sim q(\mathbf{x})}\left[\log\left(\left|\det \frac{d\Psi_{\boldsymbol{\xi}^*}(\boldsymbol{a})}{d\boldsymbol{a}}\right|\right) - \log\left(\left|\det \frac{d\Psi_*(\boldsymbol{b})}{d\boldsymbol{b}}\right|\right)\right]$$

$$(8)$$

with $\Psi_{\boldsymbol{\xi}^*}^{-1}(\mathbf{x}) = \boldsymbol{a}$ and $\Psi_*^{-1}(\mathbf{x}) = \boldsymbol{b}$.

Now, simplifying the left expression from above, we obtain

$$\mathbb{E}_{\mathbf{x} \sim q(\mathbf{x})}\left[\log\left(\frac{\prod_{i=1}^d \frac{1}{\sigma_T\sqrt{2\pi}}\exp\left(-\frac{1}{2}\frac{b_i^2}{\sigma_T^2}\right)}{\prod_{i=1}^d \frac{1}{\sigma_T\sqrt{2\pi}}\exp\left(-\frac{1}{2}\frac{a_i^2}{\sigma_T^2}\right)}\right)\right] = \mathbb{E}_{\mathbf{x} \sim q(\mathbf{x})}\left[\frac{1}{2\sigma_T^2}\sum_{i=1}^d \left(a_i^2 - b_i^2\right)\right].$$

Rewriting $a_i = b_i + \epsilon_i$ for $\epsilon_i \in \mathbb{R}$ we have

$$\mathbb{E}_{\mathbf{x} \sim q(\mathbf{x})}\left[\frac{1}{2\sigma_T^2}\sum_{i=1}^d \left(2\epsilon_i b_i + \epsilon_i^2\right)\right] = \frac{1}{\sigma_T^2}\mathbb{E}_{\mathbf{x} \sim q(\mathbf{x})}\left[\sum_{i=1}^d \epsilon_i b_i\right] + \frac{1}{2\sigma_T^2}\mathbb{E}_{\mathbf{x} \sim q(\mathbf{x})}\left[\sum_{i=1}^d \epsilon_i^2\right].$$

Since $\|\boldsymbol{a} - \boldsymbol{b}\|_2 \leq r\sigma_T$, we have that $\sum_{i=1}^d (a_i - b_i)^2 \leq r^2\sigma_T^2$ and again with $a_i = b_i + \epsilon_i$ we have that $\sum_{i=1}^d \epsilon_i^2 \leq r^2\sigma_T^2$.

Hence, we have that

$$\frac{1}{2\sigma_T^2}\mathbb{E}_{\mathbf{x} \sim q(\mathbf{x})}\left[\sum_{i=1}^d \epsilon_i^2\right] \leq \frac{1}{2\sigma_T^2}\mathbb{E}_{\mathbf{x} \sim q(\mathbf{x})}\left[r^2\sigma_T^2\right] = \frac{r^2}{2},$$

where the last equality follows from the independence of the random variables in the multivariate distribution. Moreover, applying the Cauchy-Schwarz inequality, we have:

$$
\begin{aligned}
\frac{1}{\sigma_T^2}\mathbb{E}_{\mathbf{x}\sim q(\mathbf{x})}\left[\sum_{i=1}^{d}\epsilon_i b_i\right] &\leq \frac{1}{\sigma_T^2}\mathbb{E}_{\mathbf{x}\sim q(\mathbf{x})}\left[\left(\sum_{i=1}^{d}\epsilon_i^2\right)^{1/2}\left(\sum_{i=1}^{d}b_i^2\right)^{1/2}\right] \\
&\leq \frac{1}{\sigma_T^2}\mathbb{E}_{\mathbf{x}\sim q(\mathbf{x})}\left[r\sigma_T\left(\sum_{i=1}^{d}b_i^2\right)^{1/2}\right] \\
&= \frac{r}{\sigma_T}\mathbb{E}_{\boldsymbol{b}\sim\mathcal{N}(\mathbf{0},\sigma_T^2\mathbf{I})}\left[\left(\sum_{i=1}^{d}b_i^2\right)^{1/2}\right],
\end{aligned}
\tag{9}
$$

where the second inequality follows from the definition of $r$. Since $b_i \overset{\text{i.i.d}}{\sim} \mathcal{N}(0,\sigma_T^2)$, the sum of squares follows a Chi-squared distribution scaled by $\sigma_T^2$ (i.e., $\sum_{i=1}^{d}b_i^2 \sim \sigma_T^2\chi_d^2$). Thus:

$$
\frac{r}{\sigma_T}\mathbb{E}\left[\left(\sum_{i=1}^{d}b_i^2\right)^{1/2}\right] = \frac{r}{\sigma_T}\mathbb{E}\left[\sqrt{\sigma_T^2\chi_d^2}\right] = \frac{r}{\sigma_T}\sigma_T\mathbb{E}\left[\sqrt{\chi_d^2}\right] = r\sqrt{2}\frac{\Gamma\left(\frac{d+1}{2}\right)}{\Gamma\left(\frac{d}{2}\right)}.
$$

To this end, applying Gautschi's inequality, we have:

$$
\frac{\Gamma\left(\frac{d+1}{2}\right)}{\Gamma\left(\frac{d}{2}\right)} \leq \sqrt{\frac{d+1}{2}},
$$

which gives:

$$
r\sqrt{2}\frac{\Gamma\left(\frac{d+1}{2}\right)}{\Gamma\left(\frac{d}{2}\right)} \leq r\sqrt{2}\sqrt{\frac{d+1}{2}} = r\sqrt{d+1}.
$$

Thus, the first term of Equation (8) is upper bounded by:

$$
D_{\text{KL}}(q(\mathbf{x})\parallel p_{\boldsymbol{\xi}^*}(\mathbf{x})) \leq \frac{r^2}{2} + r\sqrt{2}\frac{\Gamma\left(\frac{d+1}{2}\right)}{\Gamma\left(\frac{d}{2}\right)} < \frac{r^2}{2} + r\sqrt{d+1}.
$$

Combining with the second term of Equation (8), we get the final bound:

$$
D_{\text{KL}}(q(\mathbf{x})\parallel p_{\boldsymbol{\xi}^*}(\mathbf{x})) \leq \frac{r^2}{2} + r\sqrt{d+1} + \mathbb{E}_{\mathbf{x}\sim q(\mathbf{x})}\left[|C(\Psi_*(\mathbf{x})) - C(\Psi_{\boldsymbol{\xi}^*}(\mathbf{x}))|\right],
$$

where $C(\Psi_{\boldsymbol{\xi}^*}(\mathbf{x})) = \log|\det J_{\Psi_{\boldsymbol{\xi}^*}}(\Psi_{\boldsymbol{\xi}^*}^{-1}(\mathbf{x}))|$.

□

## A.2 EMPIRICAL MEASUREMENT OF THE BOUND.

Figure 7: Estimation of $|C(\Psi_*(\mathbf{x})) - C(\Psi_{\boldsymbol{\xi}^*}(\mathbf{x}))|$ across various $r$ values.

We can compute the first two terms of the KD bound in Theorem 1 since we know $r$ and $d$. Although it would be hard to establish a tight analytic bound for the third term, we can still empirically estimate its magnitude in practical situations.

We test the empirical effect of $r$ on the AFHQv2 (Choi et al., 2020) and FFHQ (Karras et al., 2019) datasets. We change $r$ from 10.0 down to 0.0 and approximate $|C(\Psi_*(\mathbf{x})) - C(\Psi_{\boldsymbol{\xi}^*}(\mathbf{x}))|$ by first sampling $\mathbf{b} \sim \mathcal{N}(\mathbf{0}, \sigma_T^2\mathbf{I})$. We then approximate

$$C(\Psi_*(\mathbf{x})) \approx -\log\left(\left|\det\frac{d\Psi_*(\boldsymbol{b})}{d\boldsymbol{b}}\right|\right)$$

using a teacher solver $\Psi_*(.)$, here the Uni_PC solver with 20 NFE. Next we randomly sample an $\boldsymbol{a} \in B(\boldsymbol{b}, r\sigma_T)$, to approximate

$$C(\Psi_{\boldsymbol{\xi}^*}(\mathbf{x})) \approx -\log\left(\left|\det\frac{d\Psi_{\boldsymbol{\xi}^*}(\boldsymbol{a})}{d\boldsymbol{a}}\right|\right)$$

with the student solver $\Psi_{\boldsymbol{\xi}^*}(.)$, here the Uni_PC solver with 7 NFE. Finally, we take the absolute difference

$$H(\mathbf{x}) := |C(\Psi_*(\mathbf{x})) - C(\Psi_{\boldsymbol{\xi}^*}(\mathbf{x}))| = \left|\log\left(\left|\det\frac{d\Psi_{\boldsymbol{\xi}^*}(\boldsymbol{a})}{d\boldsymbol{a}}\right|\right) - \log\left(\left|\det\frac{d\Psi_*(\boldsymbol{b})}{d\boldsymbol{b}}\right|\right)\right|.$$

Since the computation of the Jacobian is very slow, we repeat this for 100 samples and take the average to approximate $\mathbb{E}_{\mathbf{x}\sim q(\mathbf{x})}[H(\mathbf{x})]$, which is the third term of the bound in Theorem 1. Figure 7 shows how $\mathbb{E}_{\mathbf{x}\sim q(\mathbf{x})}[H(\mathbf{x})]$ evolves as $r$ changes. We observe that as $r$ decreases so does $\mathbb{E}_{\mathbf{x}\sim q(\mathbf{x})}[H(\mathbf{x})]$. Hence, we empirically verified that the overall bound in Theorem 1 tightens with smaller $r$ as desired. However, we also observe that $\mathbb{E}_{\mathbf{x}\sim q(\mathbf{x})}[H(\mathbf{x})]$ does not converge to 0 for $r \to 0$. We hypothesize that this is related to the fact that we only compute an approximation of $\mathbb{E}_{\mathbf{x}\sim q(\mathbf{x})}[H(\mathbf{x})]$ for randomly sampled $\boldsymbol{a}$ and $\boldsymbol{b}$ with a maximum L2 distance of $r$ and not for $\boldsymbol{a}$ and $\boldsymbol{b}$ for which we have that $\Psi_{\boldsymbol{\xi}^*}(\boldsymbol{a}) = \Psi_*(\boldsymbol{b})$ as assumed by the theorem.

## B  APPLYING LD3 TO DIFFERENT DOMAINS

LD3 is applicable not only to image synthesis but also to various generation tasks. We demonstrate that LD3 can be seamlessly integrated into diverse applications, including Point Cloud Generation (Luo & Hu, 2021) and Molecular Docking (Corso et al., 2022). It is important to note that our goal is not to compete with state-of-the-art models for each task. Instead, we select one representative method per task to showcase how LD3 significantly enhances performance while using a limited number of sampling steps.

### B.1  POINT CLOUD GENERATION

Point cloud generation is the process of creating 3D representations of objects or scenes using discrete points in space. This task is crucial for a wide range of applications, including 3D modeling, virtual and augmented reality, robotics, and autonomous systems. We test LD3 on a diffusion model named DMPG (Luo & Hu, 2021) trained on the *airplain* category of the ShapeNet dataset (Chang et al., 2015).

We use codebase from (Luo & Hu, 2021) to train LD3, generate samples, and evaluate the quality of these samples which allow us to evaluate the reconstruction quality of the point clouds using Chamfer Distance (CD). The generation quality is measured with minimum matching distance (MMD), the coverage score (COV), and the Jenson-Shannon divergence (JSD). We use the same reference set following the original paper.

We generate 32 noise-sample pairs to train LD3 using 100 uniform ODE steps (teacher). We train three LD3 students with NFE=$\{4, 6, 8\}$ using MSE loss function.

Table 9 shows that LD3 consistently improves point cloud generation performance when the number of steps is limited. Interestingly, when NFE=8, LD3 even surpasses DMPG teacher in CD-COV metric.

---

[1]* Code for DMPG is available at `https://github.com/luost26/diffusion-point-cloud`.

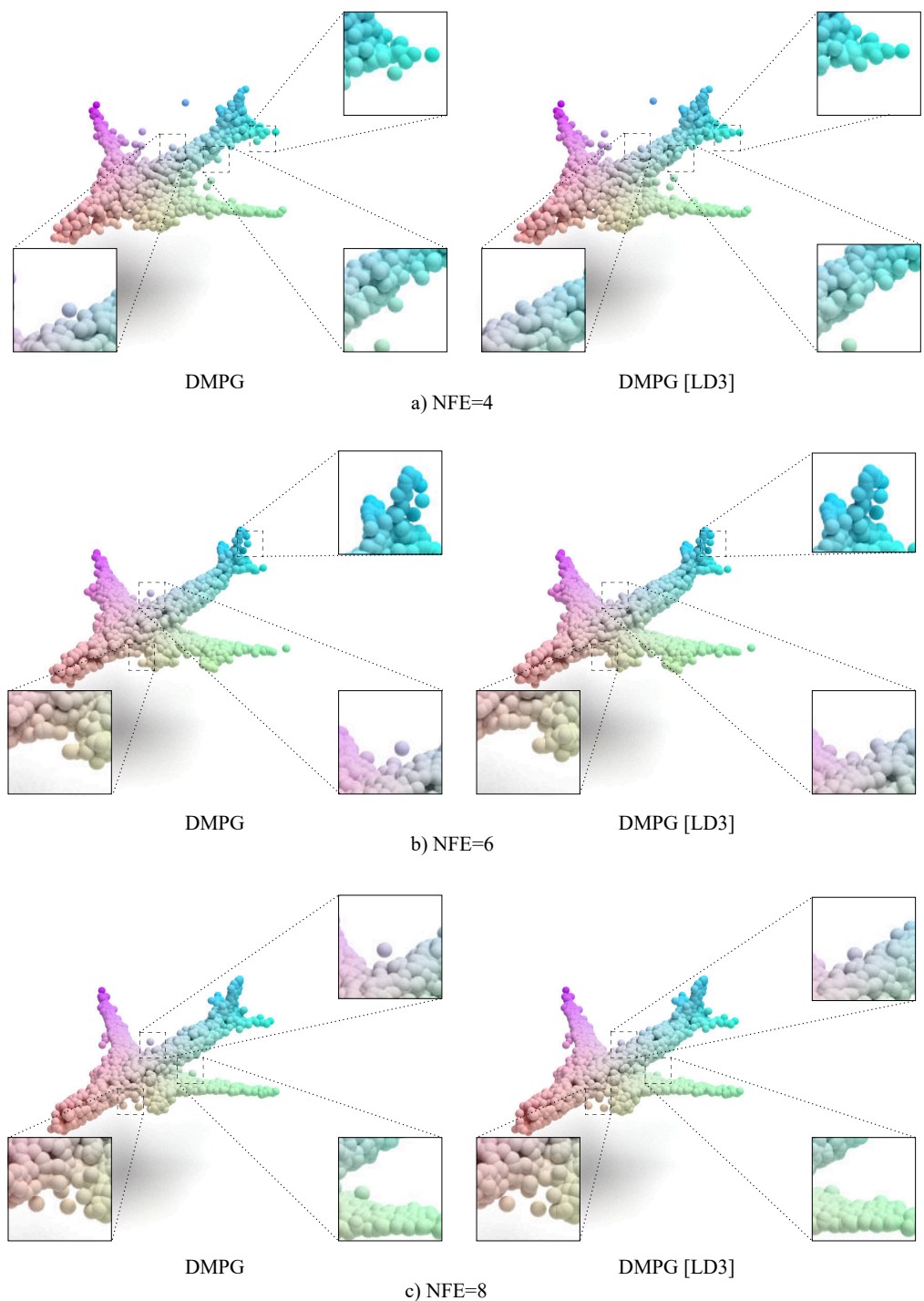

Figure 8: **Selected point cloud generation examples.** From top to bottom, we visualize student models with NFE values of 4, 6, and 8. The left column shows DMPG with the default time discretization, while the right column displays DMPG with LD3-optimized steps.

Table 9: Comparison of point cloud generation performance. We use the public code* provided by (Luo & Hu, 2021) for sampling and evaluation. CD is multiplied by $10^3$, and JSD is multiplied by $10^3$.

|  | Model DMPG | CD-MMD($\downarrow$) | CD-COV (%, $\uparrow$) | JSD ($\downarrow$) |
|---|---|---|---|---|
| | w/ Uniform | 3.82 | 34.93 | 4.08 |
| | w/ EDM | 3.50 | 42.83 | 4.38 |
| NFE=4 | w/ Quadratic | **3.42** | 46.62 | 2.68 |
| | w/ DMN | 3.44 | 45.30 | 3.04 |
| | w/ LD3 | **3.42** | **47.28** | **2.66** |
| | w/ Uniform | 3.49 | 43.66 | 2.31 |
| | w/ EDM | 3.37 | **47.28** | 2.51 |
| NFE=6 | w/ Quadratic | **3.34** | 45.80 | 1.82 |
| | w/ DMN | 3.51 | 43.16 | 2.42 |
| | w/ LD3 | **3.34** | 46.62 | **1.79** |
| | w/ Uniform | 3.39 | 46.63 | 1.71 |
| | w/ EDM | 3.35 | 45.96 | 2.02 |
| NFE=8 | w/ Quadratic | **3.34** | 46.95 | 1.63 |
| | w/ DMN | 3.37 | 46.62 | **1.61** |
| | w/ LD3 | **3.34** | **47.12** | **1.61** |
| NFE=100 | Teacher | 3.27 | 47.12 | 1.03 |

## B.2 MOLECULAR DOCKING

Molecular docking predicts how small molecules (ligands) bind to proteins, a critical process in drug discovery for identifying potential therapeutics by understanding atomic-level interactions (Fan et al., 2019). We evaluate LD3 using DiffDock (Corso et al., 2022), an advanced diffusion model that surpasses traditional docking methods. DiffDock's sampling process involves 18 steps (NFEs) of a reverse SDE. We adapt this process by converting the SDE to an ODE and by generating 10 noise-sample pairs with 18 ODE steps (NFE) to train LD3. Our student model uses 6 NFE, and we compare the performance of LD3 with that of the default uniform discretization method.

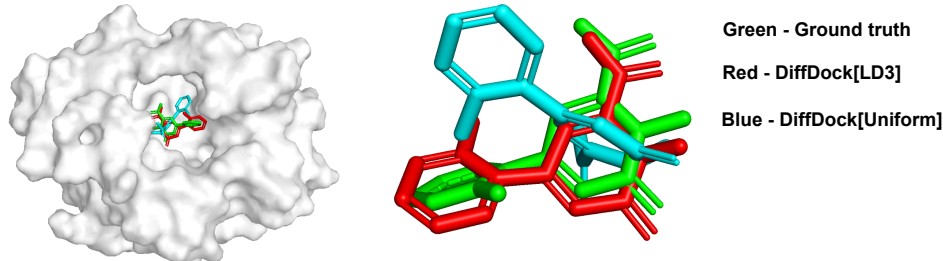

Figure 9: **Selected molecular docking example.** Comparison of predicted structures from Diff-Dock[LD3] (red) and DiffDock[Uniform] (blue) with the ground truth crystal structure (green). The structures are generated using 6 NFE. Left: Both models correctly predict the binding site. Right: DiffDock[LD3] (red) shows better alignment with the ground truth conformer (green) than DiffDock[Uniform] (blue). The RMSD values for DiffDock[LD3] and DiffDock[Uniform] are 1.70 and 3.53, respectively.

We follow the evaluation protocol of prior work Corso et al. (2022). For each protein-ligand pair, we sample 40 candidates and rank them using a pre-trained confidence model. The primary evaluation metric is the RMSD between the predicted ligand coordinates and the ground truth ligand from the PDB set. A prediction is considered accurate if its RMSD is less than 2Å (Corso et al., 2022).

As shown in Table 10, LD3 significantly narrows the gap between a 6-step student and an 18-step teacher.

Table 10: Comparison of molecular docking performance. We train LD3 and evaluate sample quality using the public code provided by (Corso et al., 2022).

| DiffDock | Top-1 RMSD: %<2Å (↑) | Top-5 RMSD: %<2Å (↑) |
|---|---|---|
| w/ DMN, NFE=6 | 33.62 | 40.40 |
| w/ EDM, NFE=6 | 33.43 | 39.66 |
| w/ Quadratic, NFE=6 | 33.43 | 39.94 |
| w/ Uniform, NFE=6 | 32.77 | 39.55 |
| w/ LD3, NFE=6 | **35.31** | **40.68** |
| Teacher, NFE=18 | 38.46 | 46.15 |

## C    COMPARISON WITH FEW-STEP GENERATION METHODS

While LD3 significantly improves generation performance in few-step regimes, it still falls short of 1-step distillation-based methods regarding generation efficiency. However, for users with access to a pre-trained diffusion model and limited computational resources, LD3 provides a practical solution for efficiently optimizing sampling performance, without having to retrain the diffusion model. LD3 is especially beneficial when a slightly slower sampling process (compared to distillation) is acceptable, as it requires only a single GPU, drastically reduces training time by hundreds of times, and maintains high efficiency.

We compare LD3 against Consistency Distillation (Song et al., 2023), Rectified Flow (Liu et al., 2022b), and Progressive Distillation (Salimans & Ho, 2022) on the CIFAR-10 dataset. Our experiment aims to answer the following question:

*"How many sampling steps, with optimized discretization, are needed to match the performance of existing few-step generation methods, and how much training effort can we save?"*

### C.1    COMPARISON WITH CONSISTENCY DISTILLATION

Table 11: Comparison with Consistency Distillation: We take the estimated training time and the number of GPUs required to train the model from (Geng et al., 2024). Both diffusion models use the NCSN++ architecture (Song et al., 2020b). The reported results for Consistency Distillation are the best-performing numbers on CIFAR-10, achieved at NFE=2. For LD3, we use the iPNDM solver.

| | Architecture | Training time | #GPUs | FID/NFE |
|---|---|---|---|---|
| CD | NCSN++ | 1 day | 8 A100s | 2.93 / NFE=2 |
| LD3 [NCSN++] | NCSN++ | < 3 minutes | 1 A100 | 15.61 / NFE=4 |
| LD3 [NCSN++] | NCSN++ | < 3 minutes | 1 A100 | 7.29/ NFE=5 |
| LD3 [NCSN++] | NCSN++ | ≈ 3 minutes | 1 A100 | 5.60 / NFE=6 |
| LD3 [NCSN++] | NCSN++ | < 6 minutes | 1 A100 | 3.19 / NFE=7 |
| LD3 [NCSN++] | NCSN++ | < 6 minutes | 1 A100 | 2.90 / NFE=8 |
| LD3 [NCSN++] | NCSN++ | < 6 minutes | 1 A100 | 3.59 / NFE=9 |
| LD3 [NCSN++] | NCSN++ | ≈ 6 minutes | 1 A100 | 2.87 / NFE=10 |

Consistency distillation (Song et al., 2023) is a diffusion-based distillation method designed to produce high-quality samples in just one or two steps. Training a consistency model on CIFAR10 requires approximately one day on 8 A100 GPUs. For a fair comparison, we train LD3 under the same conditions, using the same NCSN++ diffusion model backbone (Song et al., 2020b). With LD3, we achieve slightly better sample quality than consistency distillation when using 8 NFE. While our inference is 4 times slower, our training is 240 times faster and requires only 1/8 of the GPUs.

### C.2    COMPARISON WITH RECTIFIED FLOWS

Rectified Flow (Liu et al., 2022b) seeks to learn a straight flow from the prior distribution to the data distribution. However, the method still requires multiple steps to generate high-quality samples. For

Table 12: Comparison with Rectified Flow: We estimate the training time for Rectified Flow using the official code provided by its authors (Liu et al., 2022b), while the FID score is taken from Table 1-a of the original paper. For LD3, we employ the iPNDM solver.

|  | Architecture | Training time | #GPUs | FID/NFE |
|---|---|---|---|---|
| RF | DDPM++ | 1 day | 8 A100s | 2.58 / NFE=127 |
| LD3 [DDPM++] | DDPM++ | < 3 minutes | 1 A100 | 9.31 / NFE=4 |
| LD3 [DDPM++] | DDPM++ | < 3 minutes | 1 A100 | 8.17 / NFE=5 |
| LD3 [DDPM++] | DDPM++ | ≈ 3 minutes | 1 A100 | 3.35 / NFE=6 |
| LD3 [DDPM++] | DDPM++ | < 6 minutes | 1 A100 | 3.97 / NFE=7 |
| LD3 [DDPM++] | DDPM++ | < 6 minutes | 1 A100 | 2.81 / NFE=8 |
| LD3 [DDPM++] | DDPM++ | < 6 minutes | 1 A100 | 2.51 / NFE=9 |
| LD3 [DDPM++] | DDPM++ | ≈ 6 minutes | 1 A100 | 2.38 / NFE=10 |

instance, achieving a 2.58 FID score necessitates 127 NFE, which is comparable to LD3 applied to the same diffusion backbone but comes at a significantly higher computational cost and longer training time.

### C.3 COMPARISON WITH PROGRESSIVE DISTILLATION

Progressive Distillation (Salimans & Ho, 2022) is a distillation method that trains a student model to reduce the number of sampling steps to half of the teacher model. By repeating this process, Progressive Distillation enables the generation of high-quality images in fewer steps. While we acknowledge that the network architectures used in this comparison differ, we include this experiment to strengthen our argument that training LD3 is significantly more efficient than training a distillation-based method.

Table 13: Comparison to Progressive Distillation. The data in the first row is sourced from the original paper by Salimans & Ho (2022). Progressive Distillation introduces architectural enhancements to iDDPM (Nichol & Dhariwal, 2021).

|  | Architecture | Training time | #GPUs | FID/NFE |
|---|---|---|---|---|
| PD | iDDPM* | 1 day | 8 TPUs | 9.12 / NFE=1 |
| PD | iDDPM* | 1 day | 8 TPUs | 4.51 / NFE=2 |
| PD | iDDPM* | 1 day | 8 TPUs | 3.00 / NFE=4 |
| PD | iDDPM* | 1 day | 8 TPUs | 2.57 / NFE=8 |
| LD3 [DDPM++] | DDPM++ | < 3 minutes | 1 A100 | 9.31 / NFE=4 |
| LD3 [DDPM++] | DDPM++ | < 3 minutes | 1 A100 | 8.17 / NFE=5 |
| LD3 [DDPM++] | DDPM++ | ≈ 3 minutes | 1 A100 | 3.35 / NFE=6 |
| LD3 [DDPM++] | DDPM++ | < 6 minutes | 1 A100 | 3.97 / NFE=7 |
| LD3 [DDPM++] | DDPM++ | < 6 minutes | 1 A100 | 2.81 / NFE=8 |
| LD3 [DDPM++] | DDPM++ | < 6 minutes | 1 A100 | 2.51 / NFE=9 |
| LD3 [DDPM++] | DDPM++ | ≈ 6 minutes | 1 A100 | 2.38 / NFE=10 |

## D EXPERIMENT DETAILS

In this section, we provide more experiment details for each setting, including the codebases and the configurations for evaluation.

### D.1 PRACTICAL IMPLEMENT DETAILS

We denote $W$, $H$, and $C$ as the width, height, and number of channels of an image, respectively. Thus we have $d = C \times W \times H$. Similarly, $W'$, $H'$, and $C'$ represent the corresponding dimensions in the latent space for the Latent Diffusion model (Rombach et al., 2022), so $d = C' \times W' \times H'$.

**Teacher Solver and Data Generation.**    Generally, we select the best-performing solver at 20 NFE as our teacher solver, except for ImageNet and text-to-image generation tasks. For ImageNet, we select 10 NFE as the teacher because it performs better than higher NFE, as noted in (Sabour et al., 2024). For text-to-image generation, we also use 10 NFE, since increasing NFE does not significantly enhance our model's performance. We generate 50 to 100 samples for training. Detailed choices of solvers and NFE for the teacher solver are provided in Appendices D.1.1 to D.1.3.

**Optimizer and Trainable Parameters.**    We update three primary parameter sets during training: the step-size parameters $\boldsymbol{\xi}$, the coupling time step input to the denoising model $\boldsymbol{\xi}^c$, and the starting point $\mathbf{x}'_T$. Their optimizers are RMSprop for $\boldsymbol{\xi}$ and SGD for both $\boldsymbol{\xi}^c$ and $\mathbf{x}'_T$. The learning rates are denoted as $l_{\boldsymbol{\xi}}$, $l_{\boldsymbol{\xi}^c}$, and $l_{\mathbf{x}'_T}$. We set $l_{\boldsymbol{\xi}} = 0.005$ for pixel space datasets and $l_{\boldsymbol{\xi}} = 0.001$ for latent space datasets, while $l_{\boldsymbol{\xi}^c}$ and $l_{\mathbf{x}'_T}$ are NFE-dependent. Further details are available in Appendices D.1.1 to D.1.3.

**Initialization.**    We evaluate several baseline heuristics for $\boldsymbol{\xi}$ by computing their validation loss (a process taking only a few seconds) and choose the one that minimizes this loss for parameter initialization. We initialize $\boldsymbol{\xi}^c$ as a zero vector.

**Training.**    Initially, we freeze $\boldsymbol{\xi}^c$ and optimize only $\boldsymbol{\xi}$ and $\mathbf{x}'_T$ during the first one or two epochs (we call it phase 1). After this phase, we jointly update all parameters (we call it phase 2). At the end of each epoch, we update $\mathbf{x}'_T$ for all samples in the validation set, without updating $\boldsymbol{\xi}$ and $\mathbf{x}'_T$, with respect to $\mathcal{L}_{\text{soft}}$. We save the checkpoint that minimizes $\mathcal{L}_{\text{soft}}$ for the validation set after each training iteration.

**Evaluation.**    We evaluate our model using the FID score with 50,000 randomly generated samples. For ImageNet, we generate an equal number of samples for each class to ensure a balanced FID evaluation.

We provide more detail about our training setting in Appendices D.1.1 to D.1.3.

### D.1.1   PIXEL SPACE DIFFUSION ON CIFAR10, FFHQ, AND AFHQV2

- **Pre-trained model:**
  - EDM (Karras et al., 2022)
- **Image resolution:**
  - $W = H = 32$, $C = 3$ for CIFAR10.
  - $W = H = 64$, $C = 3$ for FFHQ and AFHQv2.
- **Training/Validation data:**
  - Size: 50/50.
  - The data are generated with the best baseline at NFE of 20, which is Uni_PC solver with Time LogSNR discretization for all three datasets.
- **Phase 1 and Phase 2:**
  - Phase 1: 2 epochs.
  - Phase 2: 5 epochs.
- **Starting point $\mathbf{x}'_T$ and $r$:**
  - SGD optimizer, $l_{\mathbf{x}'_T} = \frac{12.0}{NFE}$, momentum = 0, weight decay = 0.
  - $r = \frac{0.001 \times W \times H \times C}{NFE^2}$.
- **Time discretization parameters $\boldsymbol{\xi}$**
  - RMSprop optimizer, $l_{\boldsymbol{\xi}} = 0.005$, momentum = 0.9, weight decay = 0.
  - Gradients are clipped by the norm of $1.0$.
  - If the validation loss does not improve for $5$ consecutive iterations (patience = 5), reduce the learning rate by a factor of $0.8$. Stop decay if the learning rate reaches a minimum of $5 \times 10^{-5}$.

- **Time coupling parameters $\xi^c$:**
    - SGD optimizer, $l_{\xi^c} = \frac{0.1}{NFE}$.
    - Gradients are clipped by the norm of $1.0$.
    - If the validation loss does not improve for $5$ consecutive iterations (patience = $5$), reduce the learning rate by a factor of $0.8$. Stop decay if the learning rate reaches a minimum of $1 \times 10^{-6}$.

### D.1.2 LATENT SPACE DIFFUSION ON LSUN-BEDROOM AND IMAGENET

- **Pre-trained model:**
    - Latent Diffusion (Rombach et al., 2022)
- **Image resolution:**
    - $W = H = 256$, $C = 3$.
    - $W' = H' = 64$, $C' = 3$.
- **Guidance scale:** $2.0$ (for ImageNet).
- **Training/Validation data:**
    - Size: 50/50.
    - The data are generated with the best baseline at NFE of 20 for LSUN-Bedroom and 10 for ImageNet, using the Uni_PC solver with Time uniform discretization for LSUN-Bedroom and Time quadratic for ImageNet.
- **Phase 1 and Phase 2:**
    - Phase 1: 2 epochs.
    - Phase 2: 3 epochs.
- **Starting point $\mathbf{x}'_T$ and $r$:**
    - SGD optimizer, $l_{\mathbf{x}'_T} = \frac{12.0}{NFE}$, momentum = 0, weight decay = 0.
    - $r = \frac{0.001 \times W' \times H' \times C'}{NFE^2}$.
- **Time discretization parameters $\xi$**
    - RMSprop optimizer, $l_\xi = 0.001$, momentum = 0.9, weight decay = 0.
    - Gradients are clipped by the norm of $1.0$.
    - If the validation loss does not improve for $5$ consecutive iterations (patience = $5$), reduce the learning rate by a factor of $0.8$. Stop decay if the learning rate reaches a minimum of $5 \times 10^{-5}$.
- **Time coupling parameters $\xi^c$:**
    - SGD optimizer, $l_{\xi^c} = \frac{0.001}{NFE}$.
    - Gradients are clipped by the norm of $1.0$.
    - If the validation loss does not improve for $5$ consecutive iterations (patience = $5$), reduce the learning rate by a factor of $0.8$. Stop decay if the learning rate reaches a minimum of $1 \times 10^{-6}$.

### D.1.3 TEXT-TO-IMAGE GENERATION WITH STABLE DIFFUSION

We randomly sample 5 captions from the MS COCO dataset (Lin et al., 2015), which are:

1. *"Two individuals learning to ski along with an instructor."*
2. *"A man sitting on a chair that is on a deck over the water."*
3. *"A dog sitting at a table in front of a plate."*
4. *"Four people sit around eating food outside together."*
5. *"A cat dips its paws into a cup on a nightstand."*

We utilize Stable Diffusion v1.5 (Rombach et al., 2022) with the iPNDM solver and the NFE equal to the number of NFE of the student plus one. We used optimized GITS timestep as teacher.

- **Pre-trained model:**
  - Stable Diffusion (Rombach et al., 2022)
- **Image resolution:**
  - $W = H = 512$, $C = 3$.
  - $W' = H' = 64$, $C' = 4$.
- **Guidance scale:** $7.5$.
- **Training/Validation data:**
  - Size: 25/25.
  - The data are generated with more than one NFE compared to student. The discretization of teacher is optimized GITS.
- **Phase 1 and Phase 2:**
  - Phase 1: 2 epochs.
  - Phase 2: 3 epochs.
- **Starting point $\mathbf{x}'_T$ and $r$:**
  - SGD optimizer, $l_{\mathbf{x}'_T} = \frac{12.0}{NFE}$, momentum = 0, weight decay = 0.
  - $r = \frac{0.001 \times W' \times H' \times C'}{NFE^2}$.
- **Time discretization parameters $\boldsymbol{\xi}$**
  - RMSprop optimizer, $l_{\boldsymbol{\xi}} = 0.001$, momentum = 0.9, weight decay = 0.
  - Gradients are clipped by the norm of $1.0$.
  - If the validation loss does not improve for $5$ consecutive iterations (patience = 5), reduce the learning rate by a factor of $0.8$. Stop decay if the learning rate reaches a minimum of $5 \times 10^{-5}$.
- **Time coupling parameters $\boldsymbol{\xi}^c$:**
  - SGD optimizer, $l_{\boldsymbol{\xi}^c} = \frac{0.001}{NFE}$.
  - Gradients are clipped by the norm of $1.0$.
  - If the validation loss does not improve for $5$ consecutive iterations (patience = 5), reduce the learning rate by a factor of $0.8$. Stop decay if the learning rate reaches a minimum of $1 \times 10^{-6}$.

## D.2 BASELINE DISCRETIZATION HEURISTICS

We compare our learned time discretization with the following discretizations heuristics:

**Polynomial discretization (Time quadratic, Time uniform):** This discretization is a polynomial function. Specifically:

$$t_i = \left(\frac{i}{N}\right)^\rho (t_{\max} - t_{\min}) + t_{\min}, \quad t_{\max} = T, \ i = 0, 1, \ldots, N \tag{10}$$

here $\rho$ is often set to 1 or 2 (Song et al., 2020b; Ho et al., 2020; Lu et al., 2022b; Song et al., 2020a) which corresponds to time quadratic and time uniform discretization.

**Time EDM discretization:** First introduced by (Karras et al., 2022), Time EDM discretization has been shown to be effective with Heun's solver on EDM pre-trained model (Karras et al., 2022):

$$\sigma(t_i) = \left(\sigma_{\max}^{-\rho} + \frac{i}{N-1}\left(\sigma_{\min}^{-\rho} - \sigma_{\max}^{-\rho}\right)\right)^\rho \tag{11}$$

**Time LogSNR:** This schedule uniformly separate the logSNR. Specifically:

$$\lambda(t_i) = \frac{N-i}{N}(\lambda_{\max} - \lambda_{\min}) + \lambda_{\min}, \quad \text{where } \lambda(t_i) = \frac{\alpha(t_i)}{\sigma(t_i)} \tag{12}$$

this schedule is often used with solvers from (Lu et al., 2022a;b; Zheng et al., 2024).

### D.3    THE REMATERIALIZATION TRICK

Although the student model has few trainable parameters, the memory cost of maintaining the forward pass state scales linearly with the number of inference steps when taking gradients concerning model samples. This scaling can quickly become unfeasible given the large size of DPM architectures. To address this, we adopt the gradient rematerialization technique proposed in (Watson et al., 2022), following (Kumar et al., 2019). Rather than storing specific forward pass outputs for backward pass computations, we recompute them as needed, trading $\mathcal{O}(N)$ memory cost for $\mathcal{O}(N)$ computation time. This involves rematerializing calls to the pre-trained DDPM while keeping all progressively denoised images from the sampling chain in memory. Our model exhibits rapid convergence, even with a small batch size of 1 or 2.

## E    MORE COMPARISON WITH TIME DISCRETIZATION OPTIMIZATION METHODS

In this section, we further compare our method with some time discretization methods such as AYS (Sabour et al., 2024) and GGDM + PRED (Watson et al., 2022).

- AYS (Sabour et al., 2024) focuses on aligning the discretized sampling process with the continuous-time process in diffusion models. The authors introduce the KL-divergence Upper Bound (KLUB) objective to formalize the optimization of the sampling schedule. This objective measures the discrepancy between the discrete and continuous processes, guiding the optimization toward schedules that minimize this discrepancy. However, optimizing this objective function is challenging due to its high variance, often requiring hours of running time on multiple GPUs.

- GGDM + PRED + TIME  (Watson et al., 2022) is a Differentiable Diffusion Sampler Search method designed to improve the Kernel Inception Score by optimizing time discretization. However, their method requires 50,000 training iterations with a batch size of 512 to achieve convergence.

We compare our method with AYS in Appendix E.1 on CIFAR10 and FFHQ. Since the official implementation of AYS has not been published, we have used the numbers reported in their paper for the settings we have in common. We also compare our method with AYS and GGDM + PRED + TIME on an unconditional ImageNet-64 dataset in Appendix E.2 trained with a diffusion model from (Nichol & Dhariwal, 2021).

### E.1    COMPARE TO AYS ON CIFAR10 AND FFHQ

Table 14: Comparison of LD3 to AYS (Sabour et al., 2024) (results sourced from (Sabour et al., 2024)) and the best baseline (Best) among *time uniform*, *time quadratic*, *time EDM*, and *time LogSNR*. Unlike the experiments in Table 2, which use DPM_Solver++ (3M) (order = 3), we use DPM_Solver++ (2M) (order = 2) following (Sabour et al., 2024).

| Pre-trained DPM (Karras et al., 2022) | CIFAR10 | | FFHQ | |
|---|---|---|---|---|
| NFE | 10 | 20 | 10 | 20 |
| DPM_Solver++ (2M) (Best) | 5.07 | 2.37 | 7.07 | 3.41 |
| DPM_Solver++ (2M) (AYS) | **2.98** | **2.10** | 5.43 | 3.29 |
| DPM_Solver++ (2M) LD3 | **3.38 ± 0.64** | 2.36 ± 0.02 | **3.98 ± 0.10** | **2.89 ± 0.03** |

We compare LD3 with AYS (Sabour et al., 2024) on CIFAR10 and FFHQ. Since the implementation of AYS has not been published, we use their reported result and run LD3 on CIFAR10 and FFHQ with pre-trained EDM model (Karras et al., 2019) using DPM_Solver++ (2M). It can be seen from Table 14 that LD3 achieves competitive results with AYS. Specifically, YAS outperforms LD3 on CIFAR10 while our model achieves significantly better results on the FFHQ dataset.

## E.2 COMPARE TO AYS AND GGDM+PRED+TIME ON IMAGENET-64

Table 15: Comparison of LD3 to AYS (Sabour et al., 2024) and GGDM+PRED+TIME (Watson et al., 2022) on unconditional ImageNet-64 dataset. Baseline performance is sourced from (Sabour et al., 2024).

| Model | Solver | Discretization method | NFE=5 | NFE=10 | NFE=15 | NFE=20 | NFE=25 |
|-------|--------|----------------------|-------|--------|--------|--------|--------|
| 3M steps | DDIM | Time Uniform | 135.4 | 40.70 | 28.54 | 24.23 | 22.13 |
| | DDIM | Time Quadratic | 409.1 | 148.6 | 67.65 | 45.60 | 36.11 |
| | GGDM + PRED + TIME | Learned Schedule | **55.14** | **37.32** | **24.69** | **20.69** | **18.40** |
| 1.5M steps | DDIM | Time Uniform | 145.01 | 42.51 | 30.32 | 26.60 | 24.77 |
| | DDIM | AYS | 50.38 | 29.23 | 24.21 | 22.26 | 21.42 |
| | DDIM | LD3 | **49.79** | **28.61** | **22.93** | **21.16** | **19.81** |

Following Sabour et al. (2024), we compare our model with GGDM + PRED + TIME (Watson et al., 2022) on the ImageNet-64 dataset trained using iDDPM (Nichol & Dhariwal, 2021). The sampler being used is DDIM. Since the code for GGDM + PRED + TIME has not been published, we rely on the reported numbers in their paper. We ran our model using the published checkpoint from Nichol & Dhariwal (2021), which was trained for 1.5 million steps. Table 15 shows that, despite testing on a less trained model, LD3 significantly outperforms GGDM + PRED + TIME in the small steps regime (NFE < 20). Additionally, we compared LD3 with AYS on this dataset, and our method shows better results in all settings.

## F ADDITIONAL RESULTS

### F.1 LD3 ON PIXEL SPACE LSUN-BEDROOM-256 TRAINED WITH IDDPM

To prove the effectiveness of our method on high-resolution pixel space diffusion models, we evaluate LD3 on LSUN-Bedroom-256 dataset trained with iDDPM (Nichol & Dhariwal, 2021). We use their implemented solver, DDIM sampler, and compare the FID score generated using different discretization strategies. Table 16 compares LD3 with commonly used discretization strategies. LD3 consistently demonstrates superior performance across most NFEs (Number of Function Evaluations), achieving significantly lower FID scores than the other methods. For instance, at NFE 4, LD3 records an FID of 92.43, outperforming Time Uniform (135.57), Time Quadratic (129.88), and Time LogSNR (178.69). This trend continues as NFE increases, with LD3 maintaining the lowest FID scores up to NFE 9. However, at NFE 10, Time Uniform slightly surpasses LD3, achieving an FID of 22.62 compared to LD3's 23.03, marking the only instance where LD3 is not the top performer.

Table 16: FID comparison between LD3 and commonly used discretization strategies on LSUN-Bedroom-256, pixel space pre-trained model iDDPM (Nichol & Dhariwal, 2021), and DDIM sampler.

| Solver | Discretization | NFE | | | | | | |
|--------|---------------|-----|-----|-----|-----|-----|-----|-----|
| | | 4 | 5 | 6 | 7 | 8 | 9 | 10 |
| DDIM | Time LogSNR | 178.69 | 142.71 | 106.64 | 85.24 | 67.72 | 56.67 | 47.80 |
| | Time Uniform | 135.57 | 88.04 | 61.06 | 44.30 | 33.66 | 26.81 | **22.62** |
| | Time Quadratic | 129.88 | 101.76 | 81.18 | 65.94 | 52.77 | 45.86 | 39.47 |
| | Time EDM | 294.54 | 282.52 | 273.05 | 265.34 | 258.88 | 252.15 | 245.67 |
| | LD3 | **92.43** | **65.95** | **49.30** | **37.85** | **31.73** | **25.82** | 23.03 |

## F.2    FID Progression during training

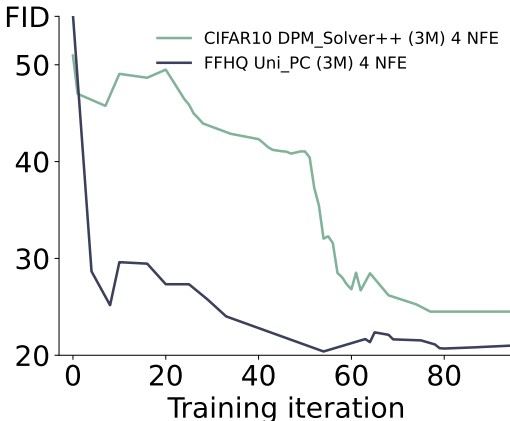

Figure 10: **FID progression during training.** We report the FID values during training for CIFAR10 and FFHQ datasets, using 4 NFE with the DPM_Solver++ (3M) for CIFAR10 and the Uni_PC (3M) solver for FFHQ. The FID, computed based on 5000 generated samples, is evaluated each time the validation loss decreases.

In our method, we save the optimal parameters based on the validation loss criterion. Figure 10 shows a decreasing trend in FID values throughout the training process. This trend suggests a correlation between FID and validation loss, indicating that FID generally decreases as the validation loss decreases. This observation further validates the effectiveness of our method.

## F.3    Stable diffusion trained with different prompt sets

To demonstrate the robustness and independence of the method with respect to the training prompts, we use different subsets of 5 prompts for training. Interestingly, despite varying the prompts, the performance remains consistent, as shown in  Table 17.

Table 17: FID score averaged over 5 runs on Stable Diffusion v1.5 using the iPNDM [LD3] solver.

|  | NFE=4 | NFE=5 | NFE=6 | NFE=7 |
|---|---|---|---|---|
| FID | $17.34 \pm 0.02$ | $13.22 \pm 0.12$ | $12.48 \pm 0.08$ | $12.04 \pm 0.22$ |

## F.4 FULL RESULT TABLES

Table 18: **Full FID comparison on CIFAR10 32×32** (Krizhevsky & Hinton, 2009).

| Solver | Discretization | NFE=4 | NFE=5 | NFE=6 | NFE=7 | NFE=8 | NFE=9 | NFE=10 |
|---|---|---|---|---|---|---|---|---|
| DPM_Solver++ (3M) | Time LogSNR | 46.59 | 24.99 | 12.16 | 6.88 | 4.62 | 3.54 | 3.08 |
| | Time Uniform | 282.94 | 263.80 | 249.75 | 238.26 | 227.79 | 217.86 | 208.34 |
| | Time Quadratic | 170.95 | 124.73 | 91.56 | 69.93 | 54.97 | 44.59 | 37.21 |
| | Time EDM | 50.91 | 32.13 | 18.64 | 11.71 | 8.44 | 6.51 | 5.16 |
| | GITS | 31.58 | 16.28 | 11.07 | 6.04 | 5.61 | 3.96 | 3.51 |
| | DMN | 39.31 | 22.88 | 12.37 | 8.21 | 7.23 | 4.35 | **2.67** |
| | LD3 | **19.39** | **9.08** | **7.51** | **5.90** | **3.42** | **2.90** | 3.09 |
| iPNDM (3M) | Time LogSNR | 35.04 | 20.52 | 11.80 | 7.69 | 5.67 | 4.37 | 3.69 |
| | Time Uniform | 266.26 | 242.94 | 229.39 | 217.13 | 205.24 | 194.72 | 185.28 |
| | Time Quadratic | 139.72 | 96.67 | 68.82 | 52.32 | 37.82 | 27.51 | 23.40 |
| | Time EDM | 29.78 | 17.35 | 9.95 | 7.61 | 5.41 | 5.00 | 3.80 |
| | GITS | 15.63 | 10.12 | 6.82 | 4.48 | 4.29 | 3.10 | 2.78 |
| | DMN | 28.09 | 16.76 | 9.24 | 6.85 | 7.68 | 4.75 | 3.31 |
| | LD3 | **9.31** | **8.17** | **3.35** | **3.97** | **2.81** | **2.51** | **2.38** |
| Uni_PC (3M) | Time LogSNR | 43.92 | 24.01 | 13.12 | 6.63 | 4.41 | 3.55 | 3.16 |
| | Time Uniform | 282.77 | 263.61 | 249.42 | 237.81 | 227.21 | 217.09 | 207.40 |
| | Time Quadratic | 164.78 | 117.28 | 85.14 | 65.25 | 51.91 | 42.48 | 35.52 |
| | Time EDM | 50.65 | 34.24 | 19.56 | 12.68 | 9.65 | 7.83 | 6.12 |
| | GITS | 25.32 | 13.79 | 11.19 | 6.80 | 5.67 | 4.40 | 3.70 |
| | DMN | 26.35 | 12.93 | 8.09 | 5.42 | 5.90 | 3.62 | **2.45** |
| | LD3 | **13.72** | **6.39** | **5.92** | **2.98** | **3.42** | **2.81** | 2.87 |

Table 19: **Full FID comparison on AFHQv2 64×64** (Choi et al., 2020).

| Solver | Discretization | NFE=4 | NFE=5 | NFE=6 | NFE=7 | NFE=8 | NFE=9 | NFE=10 |
|---|---|---|---|---|---|---|---|---|
| DPM_Solver++ (3M) | Time LogSNR | 27.82 | 17.88 | 10.72 | 6.15 | 4.28 | 3.64 | 3.19 |
| | Time Uniform | 153.20 | 143.45 | 134.46 | 125.39 | 116.13 | 107.01 | 98.25 |
| | Time Quadratic | 68.78 | 42.48 | 30.92 | 24.70 | 20.94 | 18.48 | 16.73 |
| | Time EDM | 19.65 | 13.34 | 9.64 | 7.82 | 6.54 | 4.99 | 4.11 |
| | GITS | 16.79 | 12.62 | 9.91 | 7.75 | 5.55 | 4.37 | 3.98 |
| | DMN | 37.54 | 30.09 | 18.41 | 14.66 | 9.10 | 5.94 | 3.24 |
| | LD3 | **13.86** | **9.48** | **5.47** | **4.27** | **3.63** | **2.73** | **2.68** |
| iPNDM (3M) | Time LogSNR | 23.20 | 15.63 | 9.55 | 5.98 | 4.49 | 3.75 | 3.19 |
| | Time Uniform | 159.22 | 139.47 | 125.17 | 113.68 | 102.99 | 92.57 | 82.89 |
| | Time Quadratic | 53.67 | 32.27 | 23.80 | 20.32 | 18.18 | 16.52 | 15.10 |
| | Time EDM | 15.35 | 9.01 | 6.26 | 4.73 | 3.83 | 3.55 | 3.02 |
| | GITS | 12.89 | 8.72 | 6.10 | 5.48 | 4.03 | 3.47 | 3.26 |
| | DMN | 33.15 | 26.11 | 16.01 | 13.03 | 10.12 | 6.68 | 3.22 |
| | LD3 | **9.96** | **6.09** | **3.63** | **2.97** | **2.63** | **2.25** | **2.27** |
| Uni_PC (3M) | Time LogSNR | 33.78 | 13.01 | 8.27 | 5.07 | 4.60 | 4.46 | 3.81 |
| | Time Uniform | 153.24 | 143.46 | 134.27 | 124.98 | 115.52 | 106.21 | 97.30 |
| | Time Quadratic | 66.38 | 41.56 | 30.62 | 24.26 | 20.47 | 18.04 | 16.33 |
| | Time EDM | 23.74 | 15.84 | 10.24 | 8.37 | 7.75 | 6.67 | 6.26 |
| | GITS | **12.20** | 9.45 | 7.26 | 4.85 | 3.86 | 3.26 | 2.88 |
| | DMN | 30.32 | 23.04 | 14.46 | 11.53 | 6.85 | 4.36 | 2.94 |
| | LD3 | 12.99 | **6.35** | **3.81** | **3.18** | **2.90** | **2.87** | **2.84** |

Table 20: **Full FID comparison on FFHQ 64×64 (Karras et al., 2019).**

| Solver | Discretization | NFE=4 | NFE=5 | NFE=6 | NFE=7 | NFE=8 | NFE=9 | NFE=10 |
|---|---|---|---|---|---|---|---|---|
| DPM_Solver++ (3M) | Time LogSNR | 46.14 | 22.79 | 14.01 | 8.63 | 6.18 | 4.86 | 4.18 |
| | Time Uniform | 179.07 | 165.50 | 153.56 | 142.80 | 133.07 | 124.26 | 116.30 |
| | Time Quadratic | 94.11 | 70.17 | 54.93 | 44.64 | 37.41 | 32.12 | 28.30 |
| | Time EDM | 39.59 | 23.81 | 15.29 | 11.17 | 9.67 | 8.45 | 7.01 |
| | GITS | 29.09 | 17.54 | 12.73 | 9.72 | 7.60 | 6.30 | 4.99 |
| | DMN | 40.23 | 26.31 | 14.73 | 11.21 | 10.30 | 7.37 | 4.29 |
| | LD3 | **27.99** | **13.32** | **7.53** | **4.86** | **4.42** | **3.37** | **3.55** |
| iPNDM (3M) | Time LogSNR | 36.54 | 24.57 | 16.44 | 11.31 | 8.11 | 6.39 | 5.39 |
| | Time Uniform | 39.49 | 26.15 | 20.80 | 17.52 | 14.69 | 12.50 | 10.89 |
| | Time Quadratic | 71.51 | 51.66 | 39.17 | 31.32 | 26.50 | 23.71 | 21.22 |
| | Time EDM | 29.35 | 17.52 | 11.44 | 8.76 | 6.86 | 5.96 | 4.93 |
| | GITS | 18.05 | 12.91 | 9.38 | 7.56 | 5.72 | 4.75 | 3.96 |
| | DMN | 31.30 | 20.93 | 12.12 | 10.17 | 11.00 | 8.36 | 5.24 |
| | LD3 | **17.96** | **11.98** | **6.47** | **4.88** | **3.97** | **3.57** | **3.25** |
| Uni_PC (3M) | Time LogSNR | 53.25 | 20.20 | 11.24 | 7.09 | 5.59 | 4.53 | 3.90 |
| | Time Uniform | 178.68 | 165.03 | 152.97 | 142.09 | 132.22 | 123.28 | 115.20 |
| | Time Quadratic | 89.75 | 65.94 | 51.28 | 41.47 | 34.67 | 29.85 | 26.45 |
| | Time EDM | 47.74 | 26.73 | 15.18 | 11.33 | 11.70 | 10.94 | 8.93 |
| | GITS | 21.38 | 14.33 | 12.21 | 9.94 | 7.84 | 6.19 | 4.46 |
| | DMN | 25.82 | 13.32 | 9.47 | 7.62 | 6.85 | 5.06 | 3.54 |
| | LD3 | **21.00** | **10.36** | **5.97** | **4.38** | **3.50** | **2.94** | **3.27** |

Table 21: **Full FID comparison on LSUN-Bedroom 256×256 (Yu et al., 2015).**

| Solver | Discretization | NFE=4 | NFE=5 | NFE=6 | NFE=7 |
|---|---|---|---|---|---|
| DPM_Solver++ (3M) | Time LogSNR | 80.44 | 35.81 | 16.95 | 11.38 |
| | Time Uniform | 48.82 | 18.64 | 8.50 | 5.16 |
| | Time Quadratic | 47.64 | 21.29 | 12.42 | 9.68 |
| | Time EDM | 324.41 | 294.61 | 268.96 | 243.91 |
| | GITS | 93.58 | 65.37 | 31.13 | 23.28 |
| | DMN | 35.93 | **11.13** | **4.97** | 4.21 |
| | LD3 | **28.83** | 12.17 | 5.83 | **4.16** |
| iPNDM (3M) | Time LogSNR | 55.77 | 32.51 | 20.26 | 14.52 |
| | Time Uniform | 11.93 | 6.38 | 5.08 | 4.39 |
| | Time Quadratic | 27.44 | 18.77 | 14.39 | 11.71 |
| | Time EDM | 312.44 | 284.15 | 252.37 | 221.56 |
| | GITS | 76.86 | 59.17 | 28.09 | 19.54 |
| | DMN | 11.82 | 6.15 | 4.71 | 5.16 |
| | LD3 | **8.48** | **5.93** | **4.52** | **4.31** |
| Uni_PC (3M) | Time LogSNR | 73.87 | 34.06 | 17.18 | 12.05 |
| | Time Uniform | 39.78 | 13.88 | 6.57 | 4.56 |
| | Time Quadratic | 35.97 | 15.94 | 10.91 | 9.43 |
| | Time EDM | 297.83 | 259.90 | 232.79 | 204.89 |
| | GITS | 70.93 | 47.37 | 22.33 | 17.27 |
| | DMN | 29.22 | **8.21** | **4.40** | 4.55 |
| | LD3 | **20.15** | 9.09 | 4.98 | **4.18** |

Table 22: **Full FID comparison on ImageNet 256×256 (Russakovsky et al., 2015).** We corrected a typo in the table: The results for the first 4 baselines of DPM_Solver++ and iPNDM solvers in this table were previously swapped.

| Solver | Discretization | NFE=4 | NFE=5 | NFE=6 | NFE=7 |
|---|---|---|---|---|---|
| DPM_Solver++ (3M) | Time LogSNR | 54.61 | 23.24 | 11.52 | 7.26 |
| | Time Uniform | 26.07 | 11.91 | 7.51 | 5.95 |
| | Time Quadratic | 41.94 | 22.42 | 12.04 | 7.78 |
| | Time EDM | 244.49 | 233.18 | 221.56 | 210.02 |
| | GITS | 71.77 | 51.05 | 24.72 | 13.85 |
| | DMN | 21.48 | 9.36 | 7.77 | 7.61 |
| | LD3 | **16.92** | **6.74** | **4.74** | **4.44** |
| iPNDM (3M) | Time LogSNR | 51.35 | 24.93 | 13.94 | 9.11 |
| | Time Uniform | 13.86 | 7.80 | 6.03 | 5.35 |
| | Time Quadratic | 28.54 | 15.98 | 9.50 | 6.76 |
| | Time EDM | 237.68 | 223.29 | 210.10 | 195.59 |
| | GITS | 56.00 | 43.56 | 19.33 | 10.33 |
| | DMN | 10.15 | 7.33 | 7.25 | 7.40 |
| | LD3 | **9.19** | **6.03** | **5.09** | **4.68** |
| Uni_PC (3M) | Time LogSNR | 50.26 | 19.22 | 9.08 | 5.87 |
| | Time Uniform | 20.01 | 8.51 | 5.92 | 5.20 |
| | Time Quadratic | 30.66 | 13.71 | 7.15 | 5.17 |
| | Time EDM | 235.31 | 218.15 | 203.26 | 186.88 |
| | GITS | 54.88 | 34.91 | 14.62 | 9.04 |
| | DMN | 16.72 | 7.96 | 7.54 | 7.81 |
| | LD3 | **9.89** | **5.03** | **4.46** | **4.32** |

## F.5 ADDITIONAL SAMPLES

We offer additional visual examples in Figures 11 to 23 to illustrate the qualitative effectiveness of LD3. The visual quality of LD3 surpasses discretization baselines. Our method can generate images with more visual details and less severe contrast.

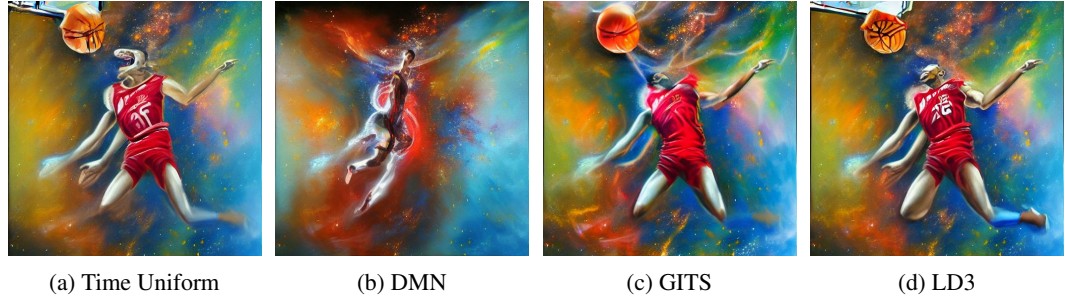

(a) Time Uniform    (b) DMN    (c) GITS    (d) LD3

Figure 11: Text prompt: *"An expressive oil painting of a basketball player dunking, depicted as an explosion of a nebula"* Stable Diffusion v1.5 (Rombach et al., 2022), iPNDM (2M) solver, NFE = 5.

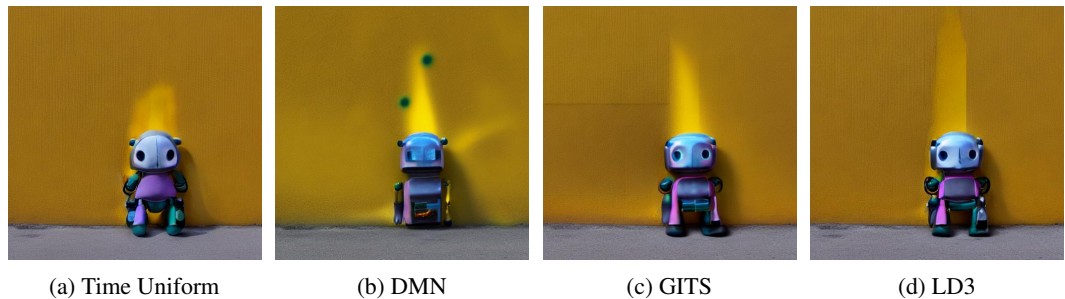

(a) Time Uniform    (b) DMN    (c) GITS    (d) LD3

Figure 12: Text prompt: *"A plush toy robot sitting against a yellow wall"* Stable Diffusion v1.5 (Rombach et al., 2022), iPNDM (2M) solver, NFE = 5.

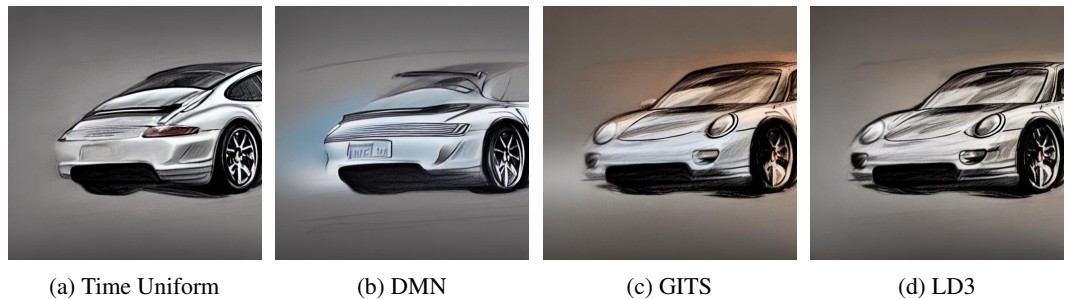

(a) Time Uniform    (b) DMN    (c) GITS    (d) LD3

Figure 13: Text prompt: *"A hand drawn sketch of a Porsche 911"* Stable Diffusion v1.5 (Rombach et al., 2022), iPNDM (2M) solver, NFE = 5.

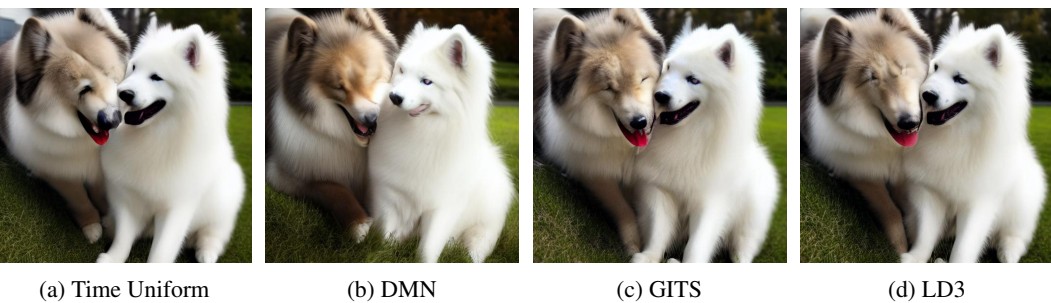

(a) Time Uniform    (b) DMN    (c) GITS    (d) LD3

Figure 14: Text prompt: *"A photo of a Samoyed dog with its tongue out hugging a white Siamese cat"* Stable Diffusion v1.5 (Rombach et al., 2022), iPNDM (2M) solver, NFE = 5.

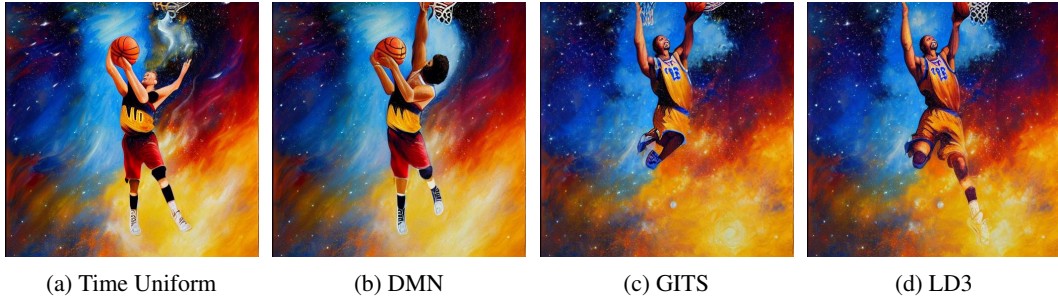

(a) Time Uniform    (b) DMN    (c) GITS    (d) LD3

Figure 15: Text prompt: *"An expressive oil painting of a basketball player dunking, depicted as an explosion of a nebula"* Stable Diffusion v1.5 (Rombach et al., 2022), iPNDM (2M) solver, NFE = 10.

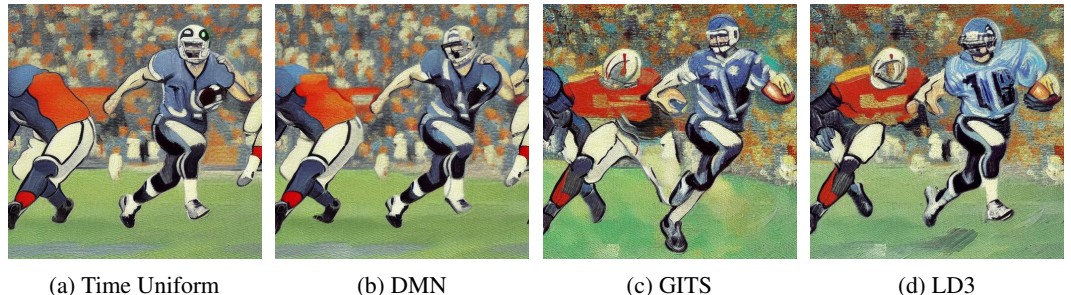

(a) Time Uniform    (b) DMN    (c) GITS    (d) LD3

Figure 16: Text prompt: *"A van Gogh style painting of an American football player"* Stable Diffusion v1.5 (Rombach et al., 2022), iPNDM (2M) solver, NFE = 10.

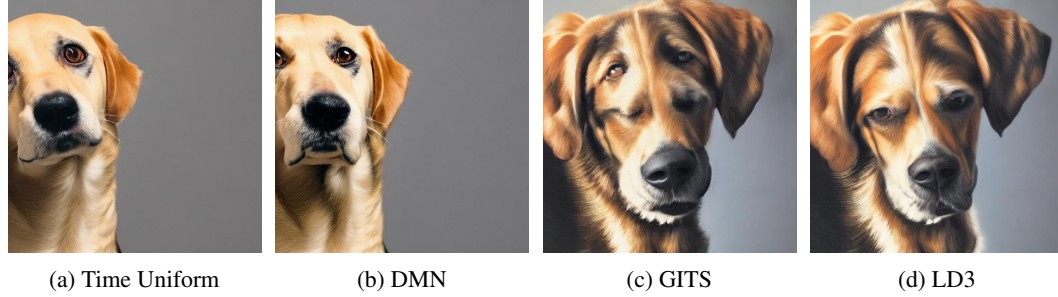

(a) Time Uniform    (b) DMN    (c) GITS    (d) LD3

Figure 17: Text prompt: *"A portrait of a dog"* Stable Diffusion v1.5 (Rombach et al., 2022), iPNDM (2M) solver, NFE = 10.

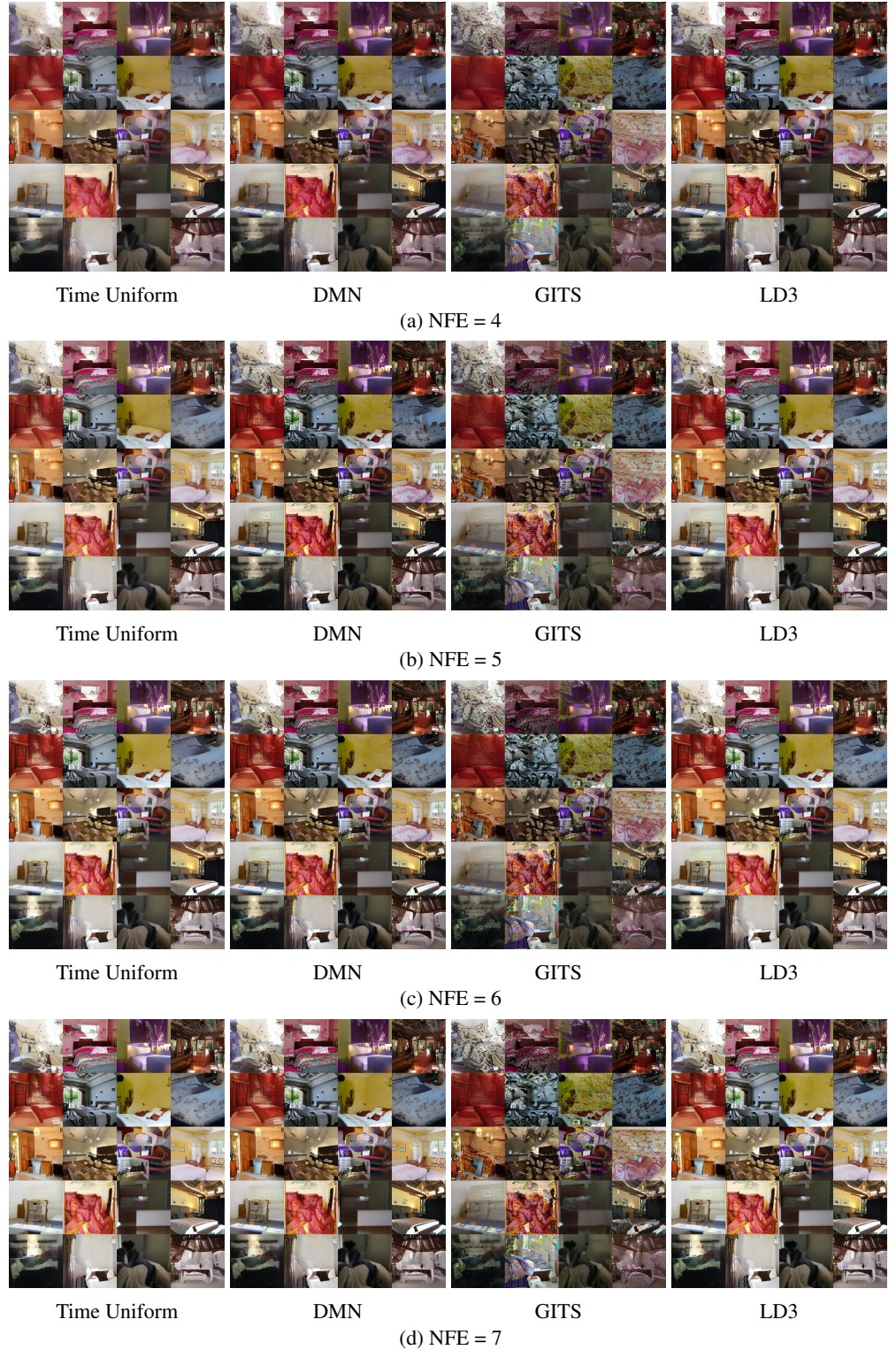

Figure 18: **Side-by-side comparison**. Random samples on LSUN-Bedroom. Qualitative comparison of methods across NFEs using the iPNDM solver, with the same randomly sampled initial noises.

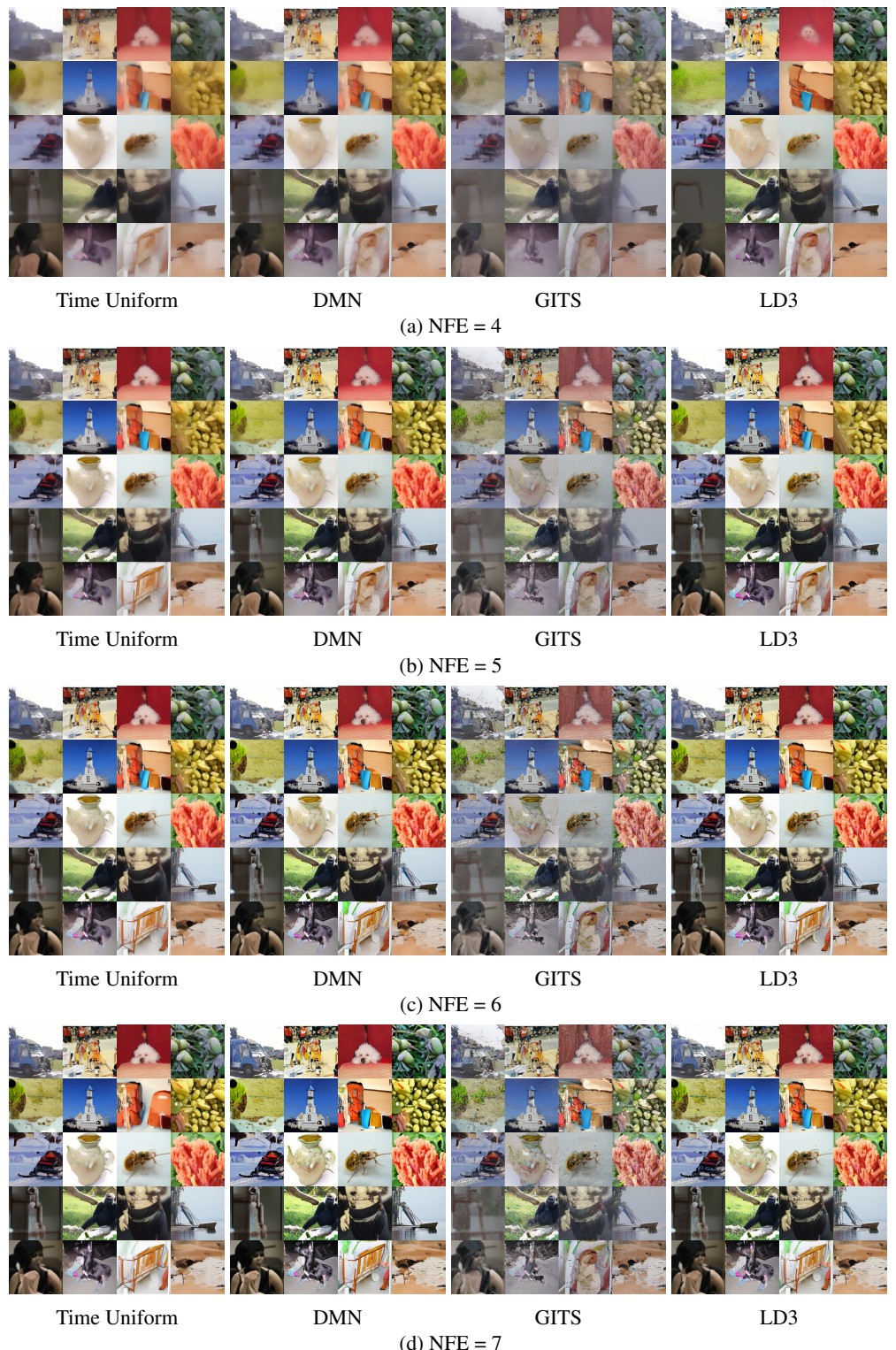

Figure 19: **Side-by-side comparison**. Random samples on class-conditional ImageNet. Qualitative comparison of methods across NFEs using the Uni_PC solver, with the same randomly sampled initial noises.

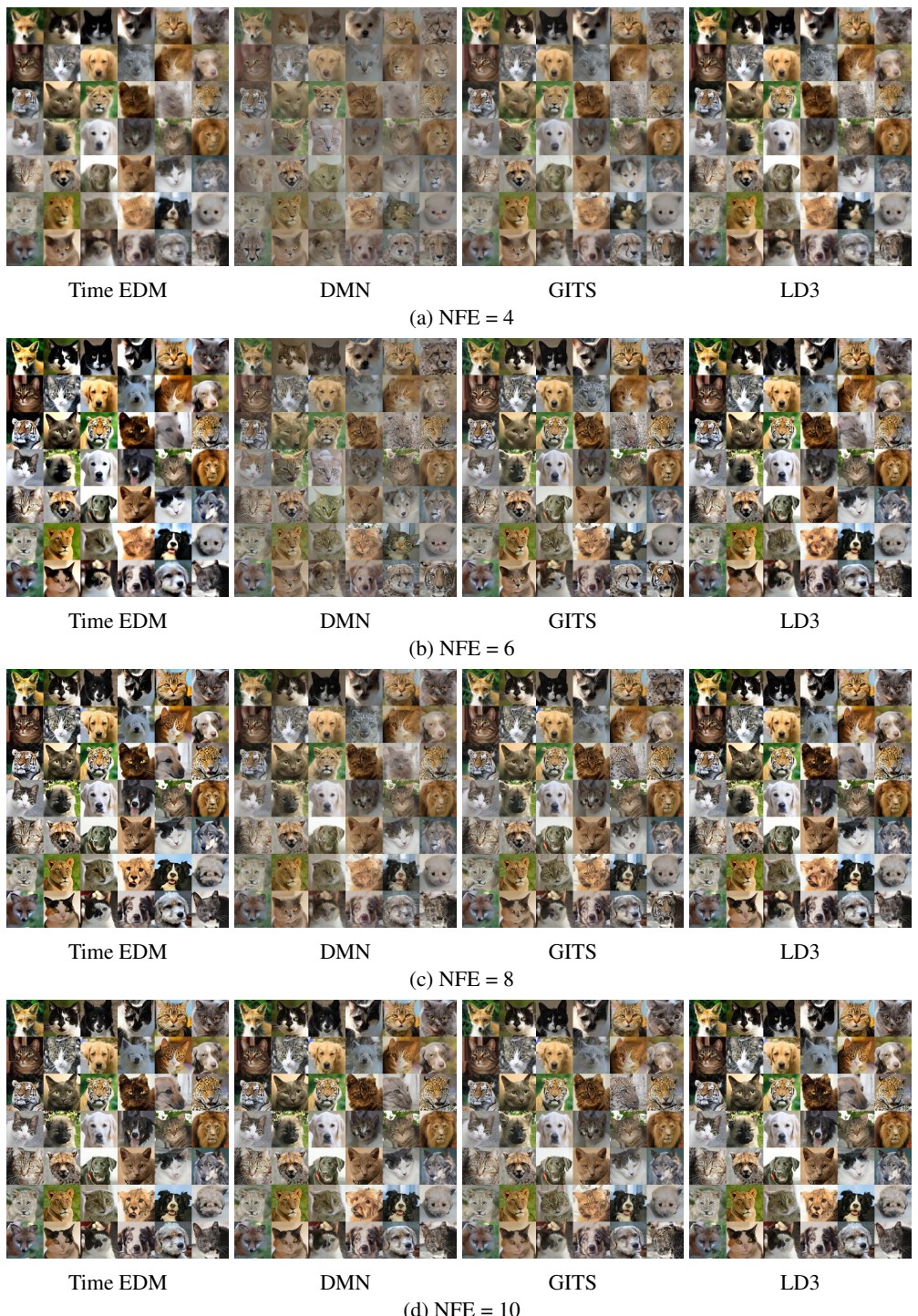

Figure 20: **Side-by-side comparison**. Random samples on AFHQv2. Qualitative comparison of methods across NFEs using the DPM_Solver++ solver, with the same randomly sampled initial noises.

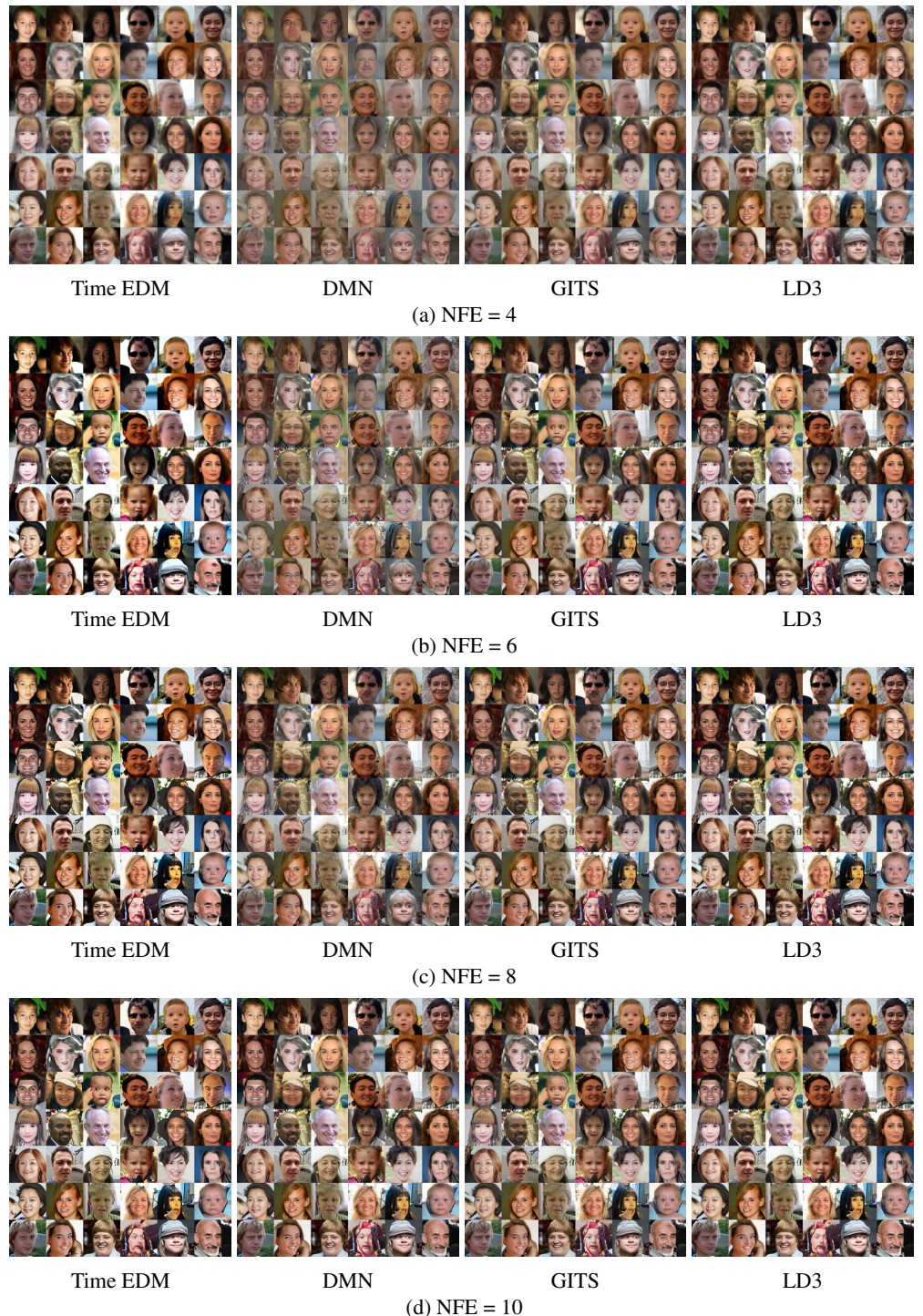

Figure 21: **Side-by-side comparison**. Random samples on FFHQ. Qualitative comparison of methods across NFEs using the DPM_Solver++ solver, with the same randomly sampled initial noises.

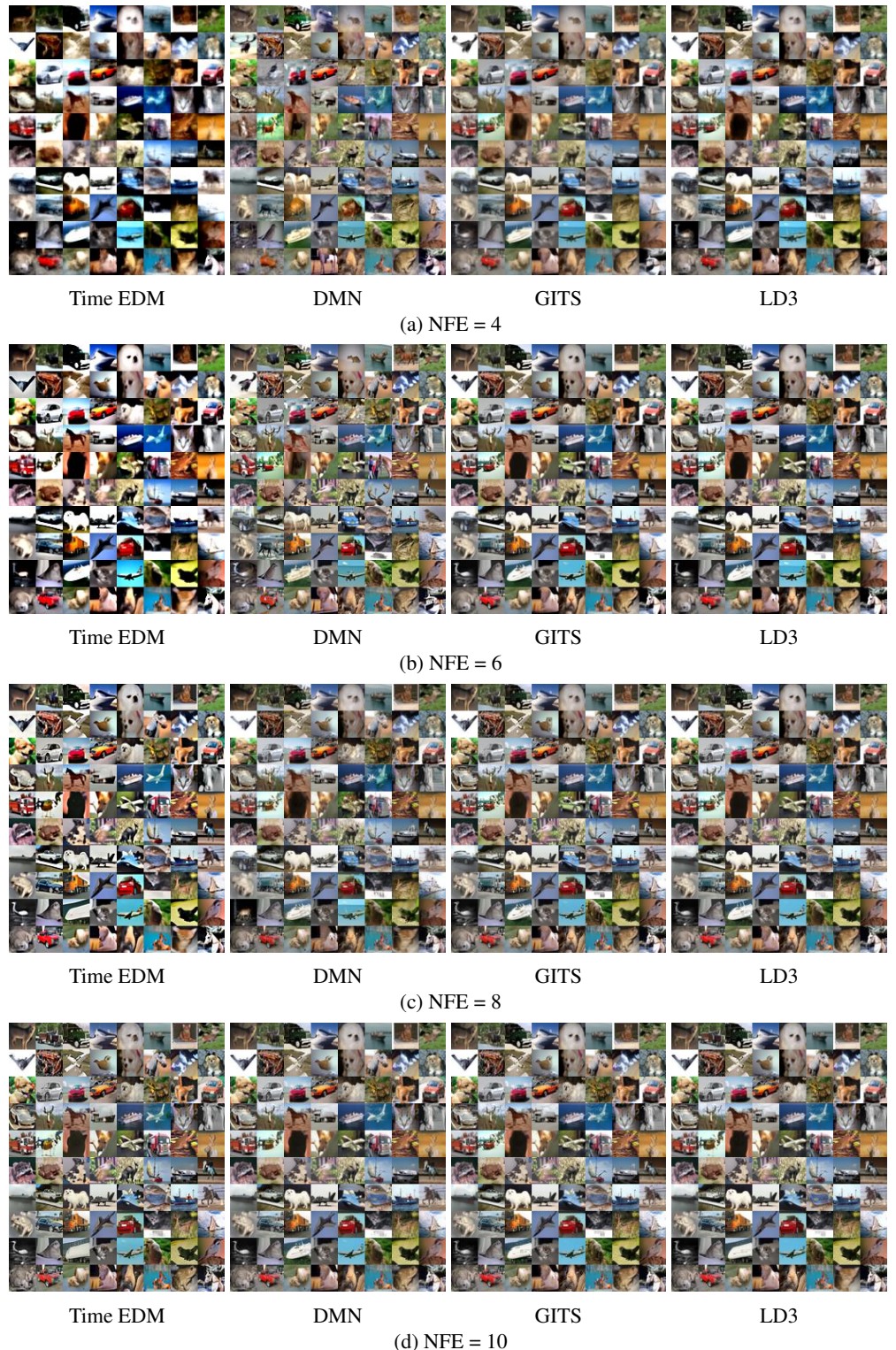

Figure 22: **Side-by-side comparison**. Random samples on CIFAR10. Qualitative comparison of methods across NFEs using the Uni_PC solver, with the same randomly sampled initial noises.

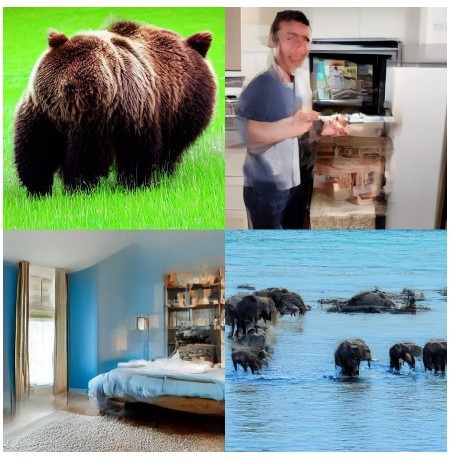 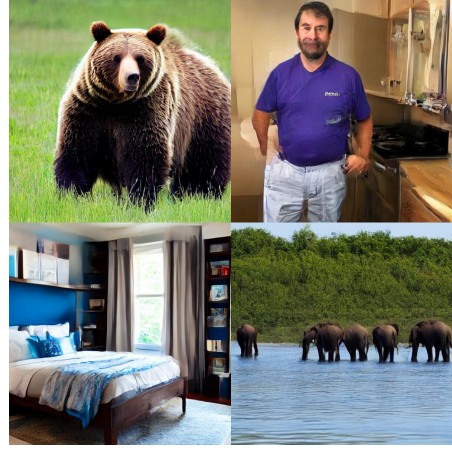

(a) Baseline, FID = 22.56            (b) LD3, FID=15.49.

Figure 23: **Side-by-side comparison**. Text prompts, from left to right, top to bottom: "A large grizzly bear with grass in the background.", "A man standing in front of a microwave, next to pots and pans.", "A bedroom scene with a bookcase, blue comforter, and window.", "A herd of elephants walking through a water-filled lake." Instaflow (Liu et al., 2023), NFE = 2.

