# OpenReview forum: "Learning to Discretize Denoising Diffusion ODEs"
_ICLR.cc/2025/Conference — ICLR 2025 Oral_

### Official Review · Reviewer_YeX6 · 2024-11-03

**Soundness:** 3
**Presentation:** 3
**Contribution:** 3
**Rating:** 8
**Confidence:** 4

**Summary:**

This paper proposes LD3, a method to discover timesteps that will yield good samples for diffusion models when the number of forward passes is very small at inference time, through optimization. The authors propose a distillation-like objective where a global notion of distance between the original model and a student model with learned timesteps is minimized. Two variants are presented, one "soft" objective achieving best results that only requires samples to be within a ball of hyperparameter radius $r$ to the teacher's samples. The soft objective additionally is linked theoretically to also upper bounding the KL divergence between the two models. Various experiments are conducted to demonstrate that LD3 achieves better FID scores than prior methods on few-step sampling, and an ablation study is included to demonstrate the importance of different components of LD3, the most important seemingly being the decoupling of timestep choice and step size, and the use of a perceptual distance (LPIPS) as opposed to pixel-based L2.

**Strengths:**

- The paper includes proofs of soundness of their proposed minimization objectives, going beyond purely empirical contribution.
- The number of experiments is substantial across both datasets, baselines from prior work, and choice of pretrained models.
- The experiments conducted include notoriously difficult datasets in the literature of diffusion step reduction like ImageNet, and shows improvement in more complex settings such as a text to image model.
- The objective is cheap to train compared to prior work; the key being that a very low batch size of 2 is permissible to use.
- Ablation studies demonstrate the importance of the proposed changes separately.
- The samples presented qualitatively look very reasonable and show clear improvement over the usual hand-crafted timestep schedules, and they fix the random seed so the same samples can be comapred.

**Weaknesses:**

There are two main areas around which the paper could be much stronger. The first is in comparisons to distillation methods, which are among the strongest in the literature. The paper includes a comparison to progressive distillation and consistency distillation in Table 9, but it is really difficult to compare these methods apples-to-apples. There are details missing (please correct me if I missed these e.g. in the supplementary material) such as what models were compared and where are the baseline scores taken from; ideally the same model should be post-trained with the different techniques. The number of forward passes across methods also doesn't match, making it difficult to draw any conclusions. One conclusion that can be drawn, however, is that progressive distillation remains better than LD3 in FID score at NFE=8, albeit requiring much more compute to distill.

The other major weakness is the lack of careful qualitative comparisons to other step reduction methods. The vast majority of the qualitative samples are compared to hand-crafted schedules, which are the weakest baselines. This is really important, especially because prior work has shown that very low FID scores can be achieved somewhat adversarially, resulting in strange samples (e.g., consider the CIFAR10 samples in the GGDM paper), so quantitative results are insufficient to truly demonstrate that LD3 improves over all prior work. Careful side-by-side comparisons of different step reduction methods, derived from the same pre-trained model and using the same initial noise and matching NFE would be significantly more convincing.

Overall, the work is strong, and the quantitative results already put this paper as a valuable contribution to the literature that should be accepted at the conference. I am opting for a weak accept, because the comparisons to distillation methods seem improper and incomplete, and the qualitative comparisons require more care. But even if so, due to the very low cost of the proposed technique and the achieved scores, the work already has intrinsic value. I strongly encourage the authors to address the concerns outlined above as it would make the work excellent.

**Questions:**

Near the introduction, the paper suggests the approach should be seen as complementary to distillation methods as opposed to going head-to-head, but the goal of both said approaches and LD3 is to achieve good sample quality with the least number of steps possible. Why do the authors argue this is the case? It is not clear to me (and it is not demonstrated in the paper) that distillation methods are compatible with optimizing the timesteps used, e.g. progressive distillation is trained to match two steps of the teacher with the student, so it's not clear a priori whether changing the timesteps later will completely break a model distilled this way.

---

> ### Author Response · Authors · 2024-11-22
> **Response to Reviewer YeX6 - Part 1**
>
> Thank you for your valuable feedback and for pointing out the areas of improvement in our manuscript.
>
> ## Weakness 1: Comparision to distillation methods
>
> We thank the reviewer for raising this question. To address this concern, we have expanded Table 9 in the main text into a separate section in Appendix-C (Comparison to Few-Step Generation Methods), where we carefully compare our method to each few-step sampling technique.
>
> In our updated comparison, we ensured that both models operate in a more similar context to enhance the validity of the comparison. While the models are not identical, we have aligned one key factor—the neural network architecture—in the case of Consistency Distillation and Rectified Flow. Although the backbone architecture differs for Progressive Distillation (PD), we believe it is still valuable to report their training cost and performance.
>
> This new setup provides a more equitable basis for comparison, highlighting each model's relative strengths and weaknesses within a consistent framework. We welcome further suggestions for improvements and thank the reviewer again for this helpful feedback.
>
> **Comparison to Consistency Distillation**
>
>
> |                  | Architecture | Training Time  | #GPUs   | FID/NFE         |
> |------------------|--------------|----------------|---------|-----------------|
> | CD               | NCSN++      | 1 day          | 8 A100s | 2.93 / NFE=2    |
> | **LD3 [NCSN++]** | **NCSN++**  | **< 6 minutes**| **1 A100** | **2.90 / NFE=8** |
>
> We reference the estimated training time and GPU requirements to train Consistency Distillation from [1]. We train LD3 with NCSN++ architecture [2] to match the same architecture used in CD. The reported results for Consistency Distillation are the best-performing numbers on CIFAR-10, achieved at NFE=2.
>
> Training a consistency model on CIFAR10 requires approximately one day on 8 A100 GPUs. With LD3, we achieve slightly better sample quality than consistency distillation when using 8 NFE. While our inference is 4 times slower, our training is 240 times faster and requires only 1/8 of the GPUs.
>
>
> **Compared to Rectified Flow**
>
> |                  | Architecture | Training Time  | #GPUs   | FID/NFE           |
> |------------------|--------------|----------------|---------|-------------------|
> | RF               | DDPM++      | 1 day          | 8 A100s | 2.58 / NFE=127    |
> | **LD3 [DDPM++]** | **DDPM++**  | **< 6 minutes**| **1 A100** | **2.51 / NFE=9**  |
>
> We estimate the training time for Rectified Flow using the official code provided by its authors [3], while the FID score is taken from Table 1-a of the original paper.
>
> Rectified Flow seeks to learn a straight flow from the prior distribution to the data distribution. However, the method still requires multiple steps to generate high-quality samples. For instance, achieving a 2.58 FID score necessitates 127 NFE, comparable to LD3 applied to the same diffusion backbone, but comes at a significantly higher computational cost and longer training time.
>
> **Compared to Progressive Distillation**
>
> |                  | Architecture  | Training Time  | #GPUs   | FID/NFE           |
> |------------------|---------------|----------------|---------|-------------------|
> | PD               | iDDPM*       | 1 day          | 8 TPUs  | 2.57 / NFE=8      |
> | **LD3 [DDPM++]** | **DDPM++**   | **< 6 minutes**| **1 A100** | **2.51 / NFE=9**  |
>
> Progressive Distillation is a method that trains a student model to reduce the number of sampling steps to half of the teacher model. By repeating this process, Progressive Distillation enables the generation of high-quality images in fewer steps. While we acknowledge that the network architectures used in this comparison are not the same, this comparison can still be a valuable source of information.
>
> All the information in the first row is taken from the original paper [4]. Progressive Distillation proposes some improvements to the architecture of iDDPM [5].
>
> [1] Geng, Z., Pokle, A., Luo, W., Lin, J. and Kolter, J.Z., 2024. Consistency Models Made Easy. arXiv preprint arXiv:2406.14548.
>
> [2] Song, Y., Sohl-Dickstein, J., Kingma, D.P., Kumar, A., Ermon, S. and Poole, B., 2020. Score-based generative modeling through stochastic differential equations. arXiv preprint arXiv:2011.13456.
>
> [3] Liu, X., Gong, C. and Liu, Q., 2022. Flow straight and fast: Learning to generate and transfer data with rectified flow. arXiv preprint arXiv:2209.03003.
>
> [4] Salimans, T. and Ho, J., 2022. Progressive distillation for fast sampling of diffusion models. arXiv preprint arXiv:2202.00512.
>
> [5] Nichol, A.Q. and Dhariwal, P., 2021, July. Improved denoising diffusion probabilistic models. In International conference on machine learning (pp. 8162-8171). PMLR.

---

> ### Author Response · Authors · 2024-11-22
> **Response to Reviewer YeX6 - Part 2**
>
> ## Weakness 2: More qualitative comparisons
>
> We thank the reviewer for the question. To address this concern, we have added more qualitative comparisons to our paper (please refer to Figure 3, 4 and Appendix F.5). We include a side-by-side comparison between LD3 and other time optimization methods, such as DMN and GITS. We use the same initial noise, matching NFE, and the same solver across all methods in these comparisons.
>
>
> ## Question: LD3 and Distillation methods
>
> We thank the reviewer for this question. The submitted paper does not mention that LD3 and distillation approaches are complementary. If we have overlooked specific sections, please let us know, and we will be happy to provide further clarification.
>
> Since both LD3 and other distillation methods aim to reduce NFE while maintaining sample quality, we have compared LD3 against Consistency Distillation, Rectified Flow, and Progressive Distillation on the CIFAR-10 dataset (see Appendix C in the revised paper). While the comparisons are not fully apples-to-apples, our goal is to address the following question:
>
> “How many sampling steps, with optimized discretization, are needed to match the performance of existing few-step generation methods, and how much training effort can we save?”
>
> In Table 6, we also show that LD3 can further enhance InstaFlow, an already optimized model for few-step generation.

---

> > ### Comment · Reviewer_YeX6 · 2024-11-25
> > **I have read the reviews**
> >
> > Thank you for the significant effort poured into the rebuttals and improving the paper. I have taken a look at the other reviews, as well as the new qualitative results and comparisons to distillation. These are major improvements that strengthen the work. Even if distillation methods can achieve similar FID with less steps, as the work clearly shows this comes at a much heftier compute price. With other applications beyond image generation and all of the above, this is a strong paper. I plan to continue recommending that this paper be accepted.
> >
> > For a camera-ready version, one remaining suggestion is to conduct some of these evaluations across NFE, rather than picking seemingly arbitrary NFE for each method where FID seems close. This will help readers understand with more transparency what are the limitations of the different methods.

---

> > > ### Author Response · Authors · 2024-11-25
> > >
> > > Dear reviewer YeX6
> > >
> > > Thank you for your positive feedback and acknowledging the improvements and quality of the paper. We will incorporate your suggestion regarding NFE evaluations in the camera-ready version.
> > >
> > > We would also politely ask you to consider raising your score so as to have it match your very positive assessment of our paper.
> > >
> > > Thank you again for your support and constructive feedback.

---

### Official Review · Reviewer_4pzZ · 2024-11-04

**Soundness:** 3
**Presentation:** 3
**Contribution:** 3
**Rating:** 8
**Confidence:** 5

**Summary:**

The paper tackles the challenge of optimizing timestep schedules for sampling in diffusion and flow-based generative models. The authors propose optimizing the schedule in order to minimize the global truncation error of multi-step sampling by reducing the discrepancy between a high-NFE teacher model and a low-NFE student model. They demonstrate that a relaxed version of this optimization problem yields even better results. Impressively, their proposed algorithm runs very quickly, achieving convergence in under 1 hour on a single GPU. A comprehensive evaluation is done using multiple pretrained diffusion models and ODE solvers. The proposed algorithm is compared against several hand-designed and learnt sampling schedules, and is shown to considerably improve the image quality in the low-NFE regime.

**Strengths:**

- The LD3 algorithm is extremely lightweight, requiring only 100 samples and less than 1 hour on a single GPU to learn optimized sampling schedules.
- The method is evaluated on a comprehensive set of pretrained models and compared against several baseline, showing improved quality in the majority of cases
- A proper ablation study is done on the various choices/hyperparameters.

**Weaknesses:**

- There are several typos in the paper. See some examples below:
    - Algorithm 1 Line 6: $x'_T ← x'_T + ...$  must be $x'_T ← x_T + ...$
    - Line 251: $x'T \rightarrow x'_T$
    - Line 251: $\Psi*(x_T) \rightarrow \Psi_*(x_T)$
- Theorem 1 requires more explanation on its invertibility assumption. Specifically, if the NFE is small, functions $\Psi_*, \Psi_\xi$ invertibility is a non-trivial fact which requires some justification on its assumption.
- The method relies on a learned perceptual distance (LPIPS) to achieve optimal results, as shown by the significant quality drop in Table 7 when switching to a standard Euclidean loss. This raises questions about how well the method might generalize to other data types beyond images.

**Questions:**

- Can the invertibility assumption in Theorem 1 be relaxed and still achieve an upper bound on the KL divergence?

---

> ### Author Response · Authors · 2024-11-22
> **Response to Reviewer 4pzZ - Part 1**
>
> Thank you for your valuable feedback and for pointing out the areas of improvement in our manuscript.
>
> ## Weakness 1: Typos in the paper.
>
> Thank you for pointing out several typos in our paper. We have corrected these errors and updated the submission accordingly. We appreciate your careful review and attention to detail.
>
> ## Weakness 2: Invertibility assumption of the theorem.
>
> Thank you for your insightful observation about the invertibility assumption. The teacher solver $\Psi_*$, which employs small step sizes, can be assumed to be invertible. The challenge lies with the student solver, especially when it operates with a small number of NFEs.
>
> The theorem bounds the KL divergence between the student distribution and teacher distribution when the loss in Equation (6) is zero. In this scenario, the student solver is already surjective since the image of $\Psi_\mathbf{\xi}$ and the image of $\Psi_*$ is identical (i.e., Im($\Psi_\mathbf{\xi}$) = Im($\Psi_*$)).
>
> To further evaluate its invertibility (i.e., ensuring it is both injective and surjective), we conducted an experiment to test the injectivity of the student solver with a small NFE (e.g., NFE=2). Specifically, we generated 1M samples from the CIFAR-10 diffusion model using the student solver with NFE=2 and checked whether any two samples were identical. We observed no such cases. While we acknowledge that this experiment is insufficient to conclude that the student solver is always injective, it provides empirical evidence of how closely our solver aligns with the invertibility assumption required by the theorem.
>
> The theorem provides a theoretical insight into the relationship between the relaxation of the requirement that the start points of the solvers are identical and the KL divergence of the student and teacher distributions. We believe it is an important addition to the paper, even if the invertibility assumption is not always met in practice.

---

> ### Author Response · Authors · 2024-11-22
> **Response to Reviewer 4pzZ - Part 2**
>
> ## Weakness 3: Beyond image applications.
>
> We thank the reviewer for raising the question about the applicability of LD3 to different tasks.
>
> We aim to demonstrate that using the L2 distance alone already enables LD3 to perform comparably or better than the best heuristic methods. For example, on the FFHQ dataset with DPM_Solver++(3M), LD3 with L2 achieves FID scores of 26.01, 16.33, and 13.47, while the best heuristic baseline achieves 46.14, 22.79, and 14.01, respectively.
>
> Additionally, LPIPS has proven to be an effective distance metric and has been utilized in distillation methods such as [1] and [2]. We acknowledge that LPIPS is specific to the domain of computer vision. Evaluating LD3’s performance on different domain tasks is an excellent approach to demonstrate its generalization capabilities and further strengthen the paper.
>
> To address this, we extended our experiments to other applications, including point cloud generation and molecular docking. Our results indicate that LD3, using the L2 loss, effectively enhances few-step generation performance in these tasks.
>
> We list the following new results in Appendix B of the updated paper. Appendix B also provides additional qualitative examples of these problems.
>
> **Application 1: Point Cloud Generation.**
> | NFE       | Model               | CD-MMD (↓) | CD-COV (%) (↑) | JSD (↓) |
> |-----------|---------------------|-------------|----------------|---------|
> | **4**     | DMPG               | 3.82        | 34.93          | 4.08    |
> |           | **DMPG [LD3]**      | **3.69**    | **38.71**      | **3.54**|
> | **6**     | DMPG               | 3.49        | 43.66          | 2.31    |
> |           | **DMPG [LD3]**      | **3.41**    | **46.29**      | **1.89**|
> | **8**     | DMPG               | 3.39        | 46.62          | 1.71    |
> |           | **DMPG [LD3]**      | **3.36**    | **47.78**      | **1.56**|
> | **100**   |  Teacher      | 3.27        | 47.12          | 1.03    |
>
> Point cloud generation creates 3D representations of objects or scenes, enabling applications in 3D modeling, AR/VR, robotics, and autonomous systems. We evaluate LD3 on a diffusion model [3] trained on the airplane category of the ShapeNet dataset [4].
>
> Using the codebase from [3], we train LD3, generate samples, and assess quality through Chamfer Distance (CD) for reconstruction and metrics like MMD, COV, and JSD for generation. LD3 is trained on 32 noise-sample pairs from a teacher with 100 uniform ODE steps, using MSE loss with student NFEs=${4, 6, 8}$.
>
> Results show LD3 improves generation performance under limited steps and, notably, outperforms the DMPG teacher in the CD-COV metric when NFE=8.
>
>
> **Application 2: Protein-Ligand Docking**
>
> | DiffDock             | Top-1 RMSD: %<2Å (↑) | Top-5 RMSD: %<2Å (↑) |
> |----------------------|-----------------------|-----------------------|
> | w/ Uniform, NFE=6   | 32.77                | 39.55                |
> | **w/ LD3, NFE=6**    | **35.31**            | **40.68**            |
> | Teacher, NFE=18      | 38.46                | 46.15                |
>
> Molecular docking predicts how small molecules bind to proteins, a key step in drug discovery. We evaluate LD3 with DiffDock [5], a diffusion model that outperforms traditional docking methods. DiffDock uses 18 steps (NFE) of reverse SDE for sampling; we adapt this by converting the SDE to an ODE and training LD3 on 10 noise-sample pairs with 18 ODE steps. Our student model uses 6 NFE, and we compare its performance to the default uniform discretization.
>
> Using the evaluation protocol from [5], we sample 40 candidates per protein-ligand pair, rank them with a pre-trained confidence model, and measure RMSD against ground truth coordinates from the PDB test set. Predictions with RMSD < 2Å are considered good. The results show that LD3 significantly reduces the performance gap between a 6-step student and an 18-step teacher.
>
> [1] Song, Y., Dhariwal, P., Chen, M. and Sutskever, I., 2023. Consistency models. arXiv preprint arXiv:2303.01469.
>
> [2] Kang, M., Zhang, R., Barnes, C., Paris, S., Kwak, S., Park, J., Shechtman, E., Zhu, J.Y. and Park, T., 2025. Distilling diffusion models into conditional gans. In European Conference on Computer Vision (pp. 428-447). Springer, Cham.
>
> [3] Luo, S. and Hu, W., 2021. Diffusion probabilistic models for 3d point cloud generation. In Proceedings of the IEEE/CVF conference on computer vision and pattern recognition (pp. 2837-2845).
>
> [4] Chang, A.X., Funkhouser, T., Guibas, L., Hanrahan, P., Huang, Q., Li, Z., Savarese, S., Savva, M., Song, S., Su, H. and Xiao, J., 2015. Shapenet: An information-rich 3d model repository. arXiv preprint arXiv:1512.03012.
>
> [5] Corso, G., Stärk, H., Jing, B., Barzilay, R. and Jaakkola, T., 2022. Diffdock: Diffusion steps, twists, and turns for molecular docking. arXiv preprint arXiv:2210.01776.

---

> > ### Comment · Area_Chair_3bhk · 2024-11-24
> > **Discussion Period Ending Soon**
> >
> > Dear Reviewer,
> >
> > The discussion period will end soon. Please take a look at the author's comments and begin a discussion.
> >
> > Thanks, Your AC

---

> ### Author Response · Authors · 2024-11-25
>
> Dear reviewer 4pzZ,
>
> Since the discussion deadline is approaching, could you kindly confirm if our response has addressed your concerns? If not, we would be happy to answer any follow-up questions and concerns.
>
> Thank you again for your constructive review!

---

> ### Comment · Reviewer_4pzZ · 2024-11-26
> **Thank you for the explanation**
>
> I appreciate the authors providing additional explanation. However, I believe the point I raised about the invertibility assumption has not been fully addressed.
> Could you clarify the experiment described further? Specifically:
> 1. How were the initial noises for the 1M samples generated? Were they IID Gaussian samples, or were they chosen to be in close proximity to each other?
> 2. Was a class-conditional model used for this experiment? And if so, were all samples generated from the same class?
> 3. What method was used to determine if two samples were identical?
>
> The additional experiments on point-clouds and protein-docking are very appreciated. However, I would like to request an expansion of the results to provide a clearer understanding of the benefits. Specifically, only a single heuristic schedule (linear schedule) is used as the baseline, yet the linear schedule is often far from the optimal choice. It would be helpful to include additional options, such as EDM's polynomial schedule, quadratic schedule, and cosine schedules as well.
>
> Since the method shows promise for use in other domains, which was one of my main concerns, I’ll be raising my score.

---

> > ### Author Response · Authors · 2024-11-26
> >
> > We thank the reviewer for pointing this out! We would like to answer your questions as follows:
> >
> > **1. How were the initial noises generated?**
> >
> > They were IID Gaussian samples.
> >
> > **2. Was a class-conditional model used for this experiment?**
> >
> > We used the unconditional CIFAR10 model.
> >
> > **3. What method was used to determine if two samples were identical?**
> >
> > We used the following procedure for the experiment
> > 1.  Initialize an empty set `unique_samples`.
> > 2. For each `sample` in the list of samples:
> > 	- Flatten `sample` and convert it to a tuple `sample_tuple`.
> > 	- If `sample_tuple` is in `unique_samples`:
> > 		- Stop the process (identical sample found).
> > 	- Else:
> > 		-  Add `sample_tuple` to `unique_samples`.
> >
> > We acknowledge that this experiment is not sufficient to conclude the invertibility of the student solver. However, even if the invertibility assumption doesn’t always hold in practice, the theorem still provides valuable theoretical insights.
> >
> > Nevertheless, the practical applications of the method are the most important part of our paper, which demonstrates strong empirical performance, not only in image generation but also in other domains such as point cloud generation and molecular docking.
> >
> > If you have any other questions or suggestions, we would be happy to discuss them to further improve the paper.

---

> > ### Author Response · Authors · 2024-11-26
> > **Thank reviewer for the positive feedback**
> >
> > Dear Reviewer,
> >
> > Thank you very much for your positive and constructive feedback.
> >
> > Regarding your suggestion to further compare the two new applications, we will do our best to address this before the deadline. If time constraints prevent us from completing it by then, we will incorporate your suggestion into the camera-ready version of the paper.
> >
> > We sincerely appreciate your valuable insights and support.

---

> > ### Author Response · Authors · 2024-12-03
> >
> > We sincerely thank Reviewer 4pzZ for the constructive feedback. In response, we have conducted additional comparisons with more baseline methods, as detailed in the tables below. In general, LD3 consistently outperforms all the baselines in both applications.
> >
> > **Protein-Ligand Docking**
> >
> >
> > | DiffDock             | Top-1 RMSD: %<2Å (↑) | Top-5 RMSD: %<2Å (↑) |
> > |----------------------|-----------------------|-----------------------|
> > | w/ DMN, NFE=6   | 33.62                | 40.40                |
> > | w/ EDM, NFE=6   | 33.43                | 39.66                |
> > | w/ Quadratic, NFE=6   | 33.43                | 39.94                |
> > | w/ Uniform, NFE=6   | 32.77                | 39.55                |
> > | **w/ LD3, NFE=6**    | **35.31**            | **40.68**            |
> > | Teacher, NFE=18      | 38.46                | 46.15                |
> >
> >
> > **Point Cloud Generation**
> >
> > We further tune our model by initializing our trainable parameters using the baselines with the smallest Wasserstein distance to the teacher and achieve better results.
> >
> > | NFE       | Model           | CD-MMD (↓) | CD-CPV (%, ↑) | JSD (↓) |
> > |-----------|-----------------|------------|---------------|---------|
> > | **4**     | Time Uniform    | 3.82       | 34.93         | 4.08    |
> > |           | Time EDM        | 3.50       | 42.83         | 4.38    |
> > |           | Time Quadratic  | **3.42**   | 46.62         | 2.68    |
> > |           | DMN             | 3.44       | 45.30         | 3.04    |
> > |           | **LD3**         | **3.42**   | **47.28**     | **2.66**|
> > | **6**     | Time Uniform    | 3.49       | 43.66         | 2.31    |
> > |           | Time EDM        | 3.37       | **47.28**     | 2.51    |
> > |           | Time Quadratic  | **3.34**   | 45.80         | 1.82    |
> > |           | DMN             | 3.51       | 43.16         | 2.42    |
> > |           | **LD3**         | **3.34**   | 46.62         | **1.79**|
> > | **8**     | Time Uniform    | 3.39       | 46.63         | 1.71    |
> > |           | Time EDM        | 3.35       | 45.96         | 2.02    |
> > |           | Time Quadratic  | **3.34**   | 46.95         | 1.63    |
> > |           | DMN             | 3.37       | 46.62         | **1.61**|
> > |           | **LD3**         | **3.34**   | **47.12**     | **1.61**|
> > | **100**   | Teacher         | 3.27       | 47.12         | 1.03    |

---

### Official Review · Reviewer_4e1X · 2024-11-04

**Soundness:** 3
**Presentation:** 3
**Contribution:** 3
**Rating:** 8
**Confidence:** 2

**Summary:**

The paper introduces a sampling method for diffusion models in order to reduce the sampling time required to generate an image. The paper proposes to learn the sampling steps from a teacher model that accurately solves the ODE by taking small step sizes. Extensive experiments show the effectiveness of the method while only requiring small amounts of training time.

**Strengths:**

- The paper is well-written and easy to follow.
- It presents an easy solution to the sampling problem of diffusion models that only requires limited training time while obtaining.
- The soft teacher loss is effective and simple to implement.
- The evaluation is thorough and includes multiple models, multiple datasets, and multiple sampling strategies.

In general, I liked the paper and I lean toward acceptance. However, since this is not my area of expertise, I would wait for the discussion with the authors and other reviewers to increase the score to Accept and recommend borderline Accept for now.

**Weaknesses:**

Although I liked the paper, there are some concerns that, if addressed, would improve the paper. In the following paragraphs, I describe my concerns in detail:

- In the table with the main results, sometimes it is not clear what the metrics are computed against. I suppose the metrics in table 2, 3, 4, and 5 are computed against random samples of the model using the accurate estimation of the ODE. However, if this metric is computed against the true distribution, the performance of the teacher with the accurate computation of the ODE should be shown (1000 steps). I think the evaluation protocol needs to be more clearly defined.

- In a similar direction, Table 6 shows the performance of a teacher model using 8 steps. Why only 8 steps are used here? Would not the teacher use a higher number of samples?

- The model used is quite simple being only composed of a single vector (or two in the decoupled version). From the results in Table 7, increasing the number of parameters leads to better results. Would increasing the complexity of the model lead to better results?

- In the limitations section I found missing that the proposed method needs to be retrained for the target number of sampling steps. One model trained to generate images with 2 samples, would not be useful for 3 and a new model would need to be trained. This might be a problem since different images might necessitate a different number of steps to achieve good quality.

**Questions:**

I would like to hear the opinion of the authors on the concerns I raised in the weaknesses section.

**Details Of Ethics Concerns:**

No ethics review.

---

> ### Author Response · Authors · 2024-11-22
> **Response to Reviewer 4e1X**
>
> Thank you for your valuable feedback and for pointing out the areas of improvement in our manuscript.
> ## Weakness 1: Evaluation protocol.
> Thank you for your thoughtful comments and for raising this question.
> The metric we use is the standard FID score (as described in lines 368 and 369) computed against the reference set from each dataset. To improve clarity, we further detailed our evaluation protocol (lines [362-364]) in the revised manuscript.
>
> We appreciate the reviewer's suggestion to add the teacher's performance to Tables 2 and 4. The number of steps of the teacher solver is often less than 20. This is because with ODE solvers, adding more NFE does not necessarily result in a performance gain (see paper [1]-Figure 2 or paper [2]-Figure 7). Appendix B - Experiment Detail already provides more details about the teacher solvers' settings.
>
> [1] Karras, T., Aittala, M., Aila, T. and Laine, S., 2022. Elucidating the design space of diffusion-based generative models. Advances in neural information processing systems, 35, pp.26565-26577.
>
> [2] Zhang, Q. and Chen, Y., 2022. Fast sampling of diffusion models with exponential integrator. arXiv preprint arXiv:2204.13902.
>
> ## Weakness 2: Number of teacher NFE in Table 6.
>
> Thank you for raising this important question. We selected the 8-step InstaFlow model because it already demonstrates strong performance with an excellent FID score on the MS-COCO dataset. To illustrate this, we evaluated InstaFlow with 10 and 20 NFEs, resulting in FID scores of 13.73 and 13.60, respectively. This indicates minimal improvement despite doubling the number of sampling steps.
> InstaFlow is specifically designed to generate high-quality samples in just a few steps. In addition to Tables 2 and 3, Table 6 shows that LD3 not only enhances standard diffusion models but also improves few-step generative models that are already optimized for efficient generation.
>
> ## Weakness 3: The complexity of LD3.
>
> We acknowledge that increasing model complexity, such as designing a more complex neural network with more parameters to predict time steps, might improve performance, as indicated by the results listed in Table 7. However, our primary goal in this work was to develop a model that learns the time discretization as simply and efficiently as possible. Efficiency is our key focus, as it aligns with the practical need for scalable methods in real-world applications. However, making the model that learns the time discretization more complex is an exciting direction for future work. Please let us know if we misunderstood your question or if we need to clarify further.
>
> ## Weakness 4: Limitation of LD3
>
> Thank you for highlighting this important point. We appreciate your suggestion and have updated the limitations section (lines [534-535]) in the revised manuscript to reflect this consideration, acknowledging it as a valuable area for future research.

---

> > ### Comment · Area_Chair_3bhk · 2024-11-24
> > **Discussion Period Ending Soon**
> >
> > Dear Reviewer,
> >
> > The discussion period will end soon. Please take a look at the author's comments and begin a discussion.
> >
> > Thanks, Your AC

---

> ### Author Response · Authors · 2024-11-25
>
> Dear reviewer 4e1X,
>
> Since the discussion deadline is approaching, could you kindly confirm if our response has addressed your concerns? If not, we would be happy to answer any follow-up questions and concerns.
>
> Thank you again for your constructive review!

---

> > ### Comment · Reviewer_4e1X · 2024-11-27
> >
> > I thank the authors for including my feedback in the new version of the paper. The author added significant changes and addressed all my main concerns. Therefore, I will increase the score of the paper to accept.

---

> > > ### Author Response · Authors · 2024-11-27
> > >
> > > Dear Reviewer 4e1X,
> > >
> > > Thank you very much for your thoughtful feedback and for recognizing our efforts in addressing your concerns. We greatly appreciate your constructive comments, which have significantly improved the quality of our paper. Your support in recommending acceptance is truly encouraging.

---

### Author Response · Authors · 2024-11-22
**Summary of Changes**

We sincerely thank all reviewers for their insightful questions and valuable feedback, which have significantly improved our paper. Based on your suggestions, we have uploaded a revised version highlighting the following main additions (marked in blue in the manuscript):

1. Additional qualitative side-by-side comparisons in Fig. 3, Fig. 4, and Appendix F.5.

2. Application of LD3 to the Point Cloud Generation and Molecular Docking tasks (Appendix B).

3. A more detailed discussion on LD3 and Few-Step generation methods (Appendix C).

4. Expanded section on limitations and broader impact (Lines 535-536).

5. Clarified evaluation protocol (Lines 362-364, and captions for Tables 2 and 4).

6. Added teacher FID scores to Tables 2 and 4.

7. Corrected typographical errors, including one on Line 6 of Algorithm 1.

Thank you again for your valuable contributions to enhancing our work.

---

### Meta-Review · Area_Chair_3bhk · 2024-12-17

**Metareview:**

The paper addresses the problem of improving the inference-time timestep schedule for  diffusion networks (in order to reduce inference time) and proposes a teacher/student framework to learn the optimal time discretization based on minimizing the KL divergence between the teacher and student's output distribution. The main claim for the paper is that this relaxed objective, compared to the standard view of strictly forcing the student to match the teacher's output given the same input, makes it easier for the student to learn this discretization process. The paper presents a thorough evaluation for several image networks and during rebuttal evaluations on point clouds and Protein-Ligand Docking. The paper also presents a theorem and demonstrates improvements in training speed, compute required. In summary, the idea behind the paper is simple, elegant, and well explained and the experimental evidence is extensive and well done.

Overall, the reviewers and I agree on the strengths of the paper (well written, strong idea, extensive evaluation with strong results). The reviewers brought up several weakness, such as lack of non-image evaluation and potentially weak baselines, that were convincingly address by the authors.

Based on the strengths and very minor weaknesses of the paper, I agree with the reviewers and advocate for acceptance. I also think that this paper merits acceptance above a poster.

**Additional Comments On Reviewer Discussion:**

Reviewers primarily focused on comparisons made by the authors. Reviewer 4pzZ asked whether the method could work on non-image settings (due to the improvement of LPIPs over L2) and the authors added several new experimental settings (point clouds and Protein-Ligand Docking) which demonstrate that the method works in other settings. Reviewer 4e1X asked about the evaluation protocol which were addressed by the authors in the rebuttal. Reviewer YeX6 asked for comparison to distillation methods and more recent, stronger step reduction methods and the authors addressed this.

Reviewer 4pzZ also asked about details regarding the theorem and the authors replied, I believe, convincingly.

---

### Decision · Program_Chairs · 2025-01-22

Accept (Oral)